# Effectively modulating thermal activated charge transport in organic semiconductors by precise potential barrier engineering

Yinan Huang[1,7], Xue Gong[2,3,7], Yancheng Meng[2,3], Zhongwu Wang[1], Xiaosong Chen[1], Jie Li[1], Deyang Ji [1,6], Zhongming Wei [4], Liqiang Li [1,5✉] & Wenping Hu [1,5]

The temperature dependence of charge transport dramatically affects and even determines the properties and applications of organic semiconductors, but is challenging to effectively modulate. Here, we develop a strategy to circumvent this challenge through precisely tuning the effective height of the potential barrier of the grain boundary (i.e., potential barrier engineering). This strategy shows that the charge transport exhibits strong temperature dependence when effective potential barrier height reaches maximum at a grain size near to twice the Debye length, and that larger or smaller grain sizes both reduce effective potential barrier height, rendering devices relatively thermostable. Significantly, through this strategy a traditional thermo-stable organic semiconductor (dinaphtho[2,3-b:2',3'-f]thieno[3,2-b]thiophene, DNTT) achieves a high thermo-sensitivity (relative current change) of 155, which is far larger than what is expected from a standard thermally-activated carrier transport. As demonstrations, we show that thermo-sensitive OFETs perform as highly sensitive temperature sensors.

[1] Tianjin Key Laboratory of Molecular Optoelectronic Sciences, Department of Chemistry, Institute of Molecular Aggregation Science, Tianjin University, 300072 Tianjin, China. [2] School of Nano-Tech and Nano-Bionics, University of Science and Technology of China, Hefei 230026 Anhui, China. [3] Suzhou Institute of Nano-Tech and Nano-Bionics (SINANO), Chinese Academy of Sciences, Suzhou 215123 Jiangsu, China. [4] State Key Laboratory of Superlattices and Microstructures, Institute of Semiconductors, Chinese Academy of Sciences, 100083 Beijing, China. [5] Joint School of National University of Singapore and Tianjin University, International Campus of Tianjin University, Binhai New City, 350207 Fuzhou, China. [6] Present address: Beijing National Laboratory for Molecular Sciences, 100190 Beijing, China. [7] These authors contributed equally: Yinan Huang, Xue Gong. ✉email: lilq@tju.edu.cn

Organic semiconductor has been demonstrated to be one highly promising material for the future-generation electronics because of light weight, low cost, mechanical flexibility, and tunable structure and property[1–4]. Electronic charge transport property of organic semiconductor materials is the basis for some applications, and usually exhibits temperature-dependent behavior[5–7]. The temperature dependence of charge transport in organic semiconductors dramatically affects and even determines their performance and applications. For example, the weak temperature dependence makes the properties of semiconductors thermo-stable, which is generally desirable for the applications of power devices and integrated circuits[8–10]. In contrast, the strong temperature dependence generates thermo-sensitive material, which may be suitable for sensing[11,12]. However, until now, the effective and reliable modulation of the temperature dependence of charge transport in organic semiconductor is still a great challenge.

Organic semiconductors generally form polycrystalline film with grain boundaries (GBs) by the commonly-used vacuum or solution preparation method[13–15]. GB, a kind of local trap, in polycrystalline film significantly affects the charge transport[16–20]. Because of the existence of traps, some charges will be trapped at GB. The trapped charges repel other charges of the same sign, which leads to the formation of space-charge region (SCR) with built-in potential[21–23]. The SCR in p-type organic semiconductor is hole depletion, and thus produces a downward local band bending (Fig. 1a), whose degree depends on the height of potential barrier ($E_B$) of SCR[24,25]. The carriers around the local bended band can hop across the potential barrier through a thermal activated way, i.e., thermionic emission[26,27]. The height of the potential barrier may be varied with bias voltage[22,28], so the effective height of the potential barrier ($E_{Be}$) plays an important role in the thermal activated charge transport. Therefore, we may modify the temperature dependence of charge transport in organic semiconductors through designing and regulating the effective height of the potential barrier. However, this strategy has not been realized previously due to the great difficulty in the precise modulation of $E_{Be}$.

Here, we have realized the precise potential barrier engineering by elaborately tuning the grain size of polycrystalline film. On this basis, we have successfully achieved the effective modulation of the temperature dependence of thermal activated charge transport in organic semiconductor. Significantly, through the above strategy, a traditional thermo-stable organic semiconductor (dinaphtho[2,3-b:2′,3′-f]thieno[3,2-b]thiophene, DNTT) has been transformed into highly thermo-sensitive when $E_{Be}$ reaches the maximum at the point of grain size ($l$) near to twice of Debye length ($l_D$) (i.e., $l \approx 2l_D$). Whereas, larger or smaller grain size both reduce $E_{Be}$, which enables these DNTT devices to be relatively thermo-stable. In addition, we also have implemented the modulation of thermal activated charge transport in pentacene by the same strategy, demonstrating the universality of this strategy. As a temperature sensor[29,30], DNTT device shows a high sensitivity (defined as the relative current change value) of 155. To our knowledge, it is the best performance among the reported temperature sensors based on organic field-effect transistor (OFET)[31–35]. As demonstrations, the DNTT OFET has achieved detection of ambient infrared (IR), identification of human touch and mapping of temperature distribution.

## Results

**Theoretical analysis of potential barrier engineering.** Figure 1a shows the energy-level diagram including band bending at GB of p-type organic semiconductor. The positive charges trapped at GB repel other charges of the same sign, and are screened by bared negative charges to remain macroscopic electrical neutrality, which results in a build-in potential $\varnothing_B$[36–40]. The SCR of hole depletion is thus formed, which produces a downward band bending of local energy level at GB, just like two back-to-back Schottky barriers[22,27,39]. The carriers at GB are severely restricted

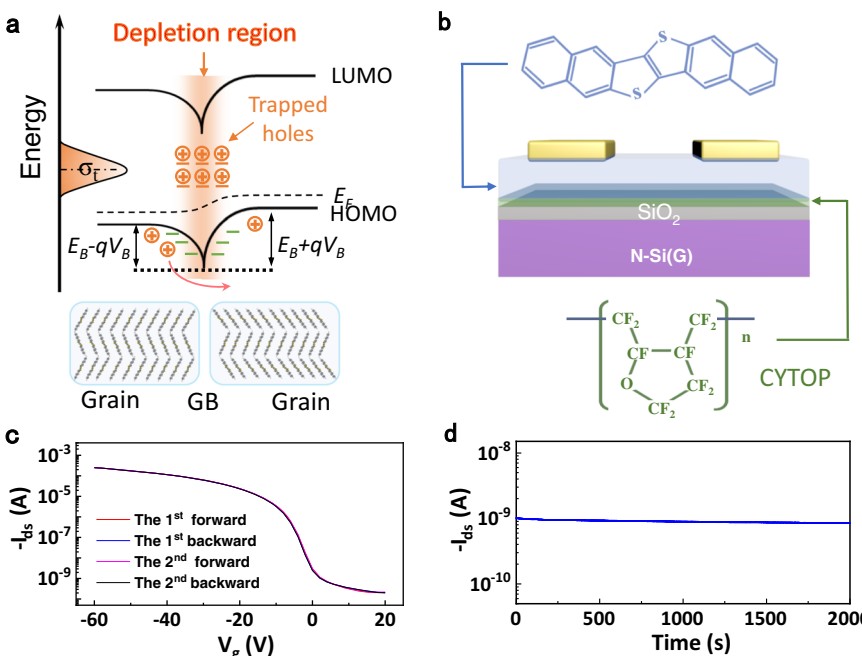

**Fig. 1 Energy band diagram, OFET structure and operational stability data. a** Schematic diagrams of energy band model at grain boundary (GB) of polycrystalline semiconductor with a bias voltage. **b** Schematic illustration of OFET and molecular structure of DNTT and CYTOP. **c** OFET transfer curves during the two cycle tests. $V_{ds} = -60$ V. **d** Bias stress curve of the OFET. $V_{gs} = 0$ V, and $V_{ds} = -60$ V. The data in (**c**, **d**) denote the excellent operational stability of DNTT OFET.

by potential barrier of SCR and basically can only hop across the barrier through thermionic emission. According to Bethe's thermionic emission theory[28,41], the net current in Schottky barrier is a superposition of forward current and reverse current. As shown in Fig. 1a, the effective height of potential barrier (mentioned as $E_{Be}$) on the forward bias side is $E_B - qV_B$, and that on the reverse bias is $E_B + qV_B$. Therefore, charge carriers on the reverse bias can hardly cross the higher potential barrier, so the corresponding current is negligible. Therefore, the current density across a GB can be given by

$$J = A^* T^2 exp\left(-\frac{E_F - E_V + E_B - qV_B}{k_B T}\right),\tag{1}$$

where $A^* = \frac{4\pi q m^* k_B^2}{h^3}$ is effective Richardson constant, $q$ is elementary charge, $m^*$ is the effective mass of the carrier, $k_B$ is the Boltzmann's constant, $h$ is Planck constant, $E_V$ is the energy level of HOMO without bending, $V_B$ is the voltage drop across a GB, and $E_B$ is the height of potential barrier at GB. $E_F - E_V$ makes no difference for devices fabricated by the same materials and processes, so $E_{Be} = E_B - qV_B$ is the key parameter that plays an important role in charge transport. These two parameters $E_B$ and $V_B$ can be given by theoretical model.

We assume that the DNTT film is composed of identical crystallites with a grain size of $l$. The minority carrier (i.e., electron) is negligible and all discussions about carriers are for majority carrier (i.e., hole). According to theoretical model of GBs[27,42], $E_B$ is dependent on the total carrier density $n$ cm$^{-3}$, trap density at GBs $Q_t$ cm$^{-2}$. Its boundary conditions can be given by the relation between grain size $l$ and Debye length $l_D$[23]. If $l < 2l_D$, the crystallite is completely depleted of carriers and the traps are partially filled (i.e., $nl < Q_t$),

$$E_B = q\emptyset_B = \frac{q^2 l^2 n}{8\varepsilon_s \varepsilon_0} \quad l < 2l_D.\tag{2}$$

If $l > 2l_D$, the crystallite is partly depleted and the traps are fully filled (i.e., $nl > Q_t$), which enable $E_B$ to be independent of $l$,

$$E_B = q\emptyset_B = \frac{q^2 Q_t^2}{8\varepsilon_s \varepsilon_0 n} \quad l > 2l_D,\tag{3}$$

where $\varepsilon_0$ and $\varepsilon_s$ are vacuum permittivity and relative permittivity, respectively. $l_D = \sqrt{\varepsilon_s \varepsilon_0 k_B T / q^2 n}$ is the characteristic length of SCR and screen length against the trapped charges at the GBs[43,44].

The resistance of a polycrystalline material consists of the contributions from the grain-boundary region ($R_B$) and the bulk of the crystallite ($R_C$). Assuming that the grains are seamless connection, the channel (with length of $L$) contains $L/l$ grains and GBs, and then $V_B$ can be described as $V_{ds}\frac{R_B}{R_B+R_C}\frac{l}{L}$, where $V_{ds}$ is the voltage drop of the channel. If we define $\beta = \frac{R_B}{R_B+R_C}$, $V_B$ becomes $\frac{\beta V_{ds}}{L}l$. Using Eqs. (2), (3) and $V_B = \frac{\beta V_{ds}}{L}l$ together, the effective height of barrier $E_{Be}$ can be expressed as follow

$$E_{Be} = E_B - qV_B = \frac{q^2 n}{8\varepsilon_s \varepsilon_0}l^2 - \frac{q\beta V_{DS}}{L}l, \, l < 2l_D.\tag{4}$$

$$E_{Be} = E_B - qV_B = \frac{q^2 Q_t^2}{8\varepsilon_s \varepsilon_0 n} - \frac{q\beta V_{DS}}{L}l, \, l > 2l_D.\tag{5}$$

On the basis of the above equations and analysis, the precise design of grain size $l$ in a certain range can tune the effective height of barrier $E_{Be}$ by controlling $E_B$ and $V_B$, which can thus modulate temperature dependence of charge transport according to Eq. (1). However, the above theoretical analysis has not been realized in organic semiconductors.

**Fabrication and characterization of OFETs**. To prove the feasibility and efficiency of potential barrier engineering for modulating thermal activated charge transport, it would be desirable to transform the thermo-stable semiconductor to be thermo-sensitive. It has been widely reported that dinaphtho[2,3-b:2′,3′-f]thieno[3,2-b]thiophene (DNTT) is a good thermo-stable field-effect semiconductor and the charge transport property barely changes from 293 K to 363 K in the OFET[45,46]. Therefore, DNTT is one kind of good model compound for this study.

To test the charge transport property of DNTT and its temperature dependence, the DNTT OFETs were prepared as shown in Fig. 1b. The perfluoro(1-butenyl vinyl ether) polymer (CYTOP) as modification layer between the semiconductor and SiO$_2$/Si was used to reduce the interfacial traps introduced by some polar groups and semiconductor disorder[47,48]. The active layer was fabricated by vacuum evaporation using multiple purified high quality DNTT, which renders the low density of impurities in the polycrystalline semiconductor film. Subsequently, the Au electrodes were deposited on the top of the DNTT polycrystalline film. All these processes aim to reduce the density of various traps except GBs as low as possible, and to avoid introducing the traps caused by these defects in the band gap. Through these strategies, DNTT OFETs exhibit standard and steady field-effect transfer characteristics without visible hysteresis during the cycle tests (Fig. 1c), and source-drain current almost remains constant under long-term bias condition (Fig. 1d). These two experimental data indicate that the devices show good operational stability without bias stress, which is the prerequisite for obtaining authentic temperature dependence of charge transport property as well as for the application in temperature sensors. In addition, the negligible bias stress indicates that there are few deep traps that may cause the bias stress.

**Temperature dependence of charge transport of DNTT films**. To testify the effect of grain size on the temperature dependence of charge transport, five polycrystalline DNTT films with different grain sizes were fabricated through elaborately regulating the process of deposition and growth of DNTT molecules (including evaporation rate, growth time and deposition temperature, for details, see Methods). Atomic force microscope (AFM) was used to characterize the morphology of the films and measure the average grain size $l$ (Supplementary Fig. 1a–e). The distribution of grain sizes in the five films are in the range of 200–520 nm. Subsequently, the temperature dependence of charge transport of the OFETs with these DNTT films was respectively investigated by measuring transfer characteristic and output characteristic curves in the temperature range of 298–358 K (for details, see Methods). The typical characteristic curves of OFETs with grain sizes of 200 nm (minimum), 350 nm (middle) and 520 nm (maximum) are shown in Fig. 2, and that of OFETs with grain sizes of 260 nm and 420 nm are shown in Supplementary Figs. 1, 2. The transfer curves at variable temperature (Fig. 2a–c, Supplementary Fig. 1g, i) and output curves (Fig. 2d–f, Supplementary Fig. 2g, i) at fixed temperature of these devices display good field-effect characteristic, indicating that the OFETs sustain the steady working status in this temperature range. The cycle-scan transfer curves at variable temperature exhibit slight hysteresis (Supplementary Fig. 3), which indicates the negligible bias stress and good operation stability, and guarantees the credibility of the investigation of thermal activated charge transport.

The current of these transfer curves increases with the increasing temperature, revealing that thermal activated behavior dominates charge transport of the polycrystalline DNTT OFETs[38,49–51]. The statistical relationship between grain size and the relative current change {[I($V_g$, T) − I($V_g$, T$_0$)]/I($V_g$, T$_0$)],

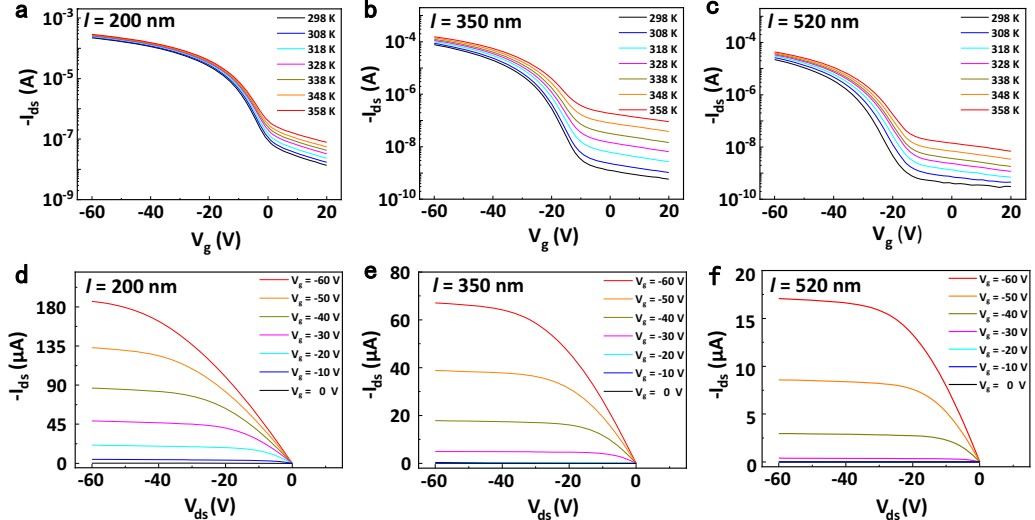

**Fig. 2 The characteristic curves of the DNTT OFETs with different grain sizes. a–c** The temperature dependent transfer curves of OFETs with grain sizes of about 200 nm, 350 nm and 520 nm. $V_{ds} = -60$ V. **d–f** The output curves of OFETs with different DNTT grain sizes corresponding (**a–c**). The character *l* in all figures denotes grain size. All electrical measurements were performed in dark ambient unless otherwise specified.

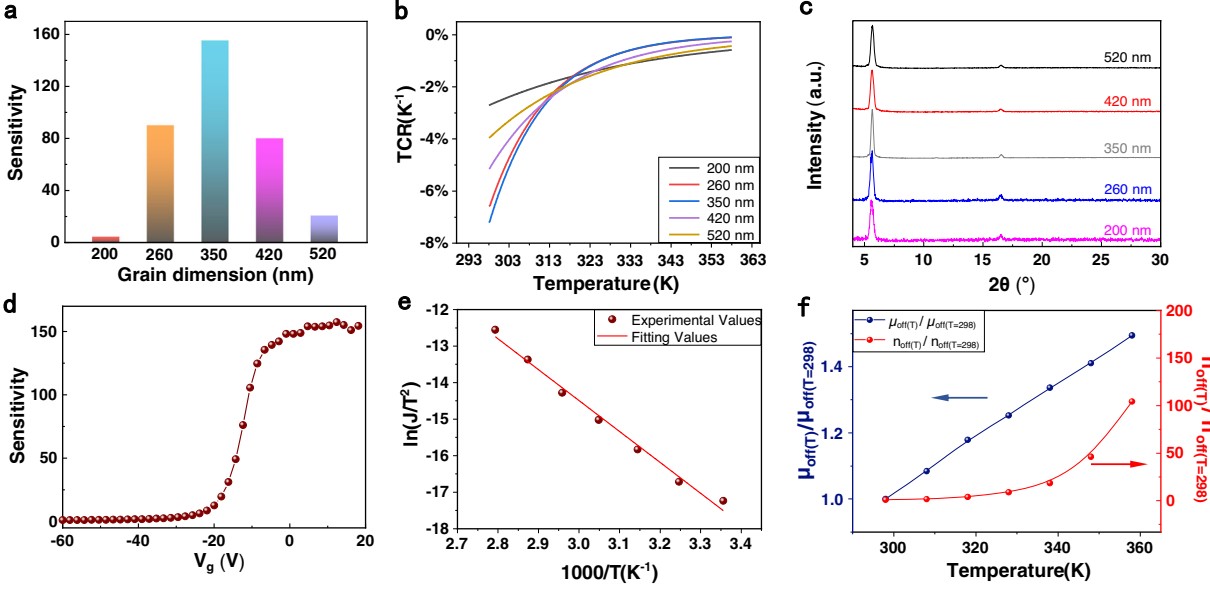

**Fig. 3 Effect of the grain size on temperature dependence of OFETs. a** Bar chart of the relationship between sensitivity and grain sizes. Sensitivity is defined as $[I(V_g, T) - I(V_g, T_0)]/I(V_g, T_0)]$, where $V_g = 20$ V, $T_0 = 298$ K, T = 358 K. **b** Temperature coefficient of resistance (TCR) of the OFETs with different grain sizes. **c** X-ray diffraction patterns of DNTT polycrystalline films with different grain sizes. **d** The correlation between the sensitivity (measured when $V_{ds} = -60$ V) and the gate voltage of the OFET. **e** The fitting plot of $\ln \frac{J}{T^2}$ as a function of $1000/T$. **f** The relative change rate of mobility and carrier density in the off-state of the OFET. The curves of (**d–f**) were measured by the OFET with grain size of 350 nm. The data of (**a**, **b** and **e–f**) were measured in the off-state ($V_g$ is 20 V) when $V_{ds}$ is −60 V.

defined as sensitivity in this work} in the off-state ($V_g$ is 20 V) from 298 to 358 K in the transfer curves is summarized in Fig. 3a and Supplementary Table 1. As expected, the devices show significant difference: the devices with the grain size distributed between 260 nm and 420 nm exhibit very strong temperature-dependent characteristic. The sensitivity of these devices is about two orders of magnitude, and the highest value is more than two and a half orders of magnitude at the grain size of about 350 nm, which value is significantly larger than the standard thermal-activated carrier transport[12,31–35]. If the typical activation energy of a standard thermally-activated carrier transport is in the range of 100–200 meV for intrinsic organic semiconductors[12,31], the expected temperature sensitivity is only around 1–3 from 298 K to

358 K. On the contrary, the electrical properties of the devices with the grain size <260 nm or larger than 420 nm are lightly changed with the varied temperature. For easy comparison with other kinds of thermosensitive materials or devices, we also calculated the temperature coefficient of resistance (TCR) as sensitivity (for details, see Supplementary Note 4). The TCR curves of the OFETs are shown in Fig. 3b, which is higher than, at least compared to, most of conventional thermosensitive materials. For example, TCR for most metals and other conductive materials is between 0.1 and 1% K[−1] [52–54], and most oxide ceramics[55,56] and silicon-based electronics[57–62] show the TCR value of 1–4% K[−1] at room temperature. This fact indicates that the DNTT OFETs with proper grain size achieve high thermo-sensitivity.

X-ray diffraction patterns of these five films (Fig. 3c) reveal that all these DNTT films have similar crystallinity, suggesting that the different temperature dependence of charge transport does not stem from the molecular ordering. The results show that well designing the grain size successfully achieves the remarkable modulation of thermal activated charge transport in organic semiconductors.

Furthermore, in single OFET, the temperature dependence of charge transport can be further tuned by the gate voltage. As shown in Fig. 3d, sensitivity shows the tiny change in the off-state (0–20 V) and on-state (−20 to −60 V), but decreases significantly in subthreshold region. Generally, the larger gate voltage leads to smaller sensitivity, which is due to the fact that the vast charges induced by gate voltage screen the potential of GBs (as expressed in Eq. (3)). The tunable sensitivity is very useful for the practical applications in the complicated environment[63].

**Modulating mechanism of the temperature dependence of DNTT polycrystalline film**. To elucidate the mechanism of the thermal activated charge transport in the polycrystalline organic semiconductor, the electrical property of the DNTT OFET at different temperature was examined. If charge carrier is mainly restrained by the potential barrier of GBs, current density should obey Eq. (1). Dividing both sides by $T^2$ and taking the logarithm, Eq. (1) becomes

$$\ln \frac{J}{T^2} = lnA^* - \left( \frac{E_F - E_V + E_B - qV_B}{k_B} \right) \frac{1}{T} \quad (6)$$

which suggests $\ln \frac{J}{T^2} \propto \frac{1}{T}$. Then, we draw the plot of $\ln \frac{J}{T^2}$ as a function of $1000/T$ for the device with grain size of 350 nm in Fig. 3e (others are shown in Supplementary Fig. 4). As expected, the curve shows good linearity, indicating that the charge transport in the DNTT film accords with the thermionic emission

model. The critical parameter $\frac{E_F - E_C + E_B - qV_B}{k_B}$ can be obtained from the slope of these fitting curves. In addition, these results also show that steeper slope indicates higher temperature sensitivity (i.e., 350 nm > 520 nm > 200 nm), which will be discussed detailly in the Discussion Section below. Moreover, the relative change of field-effect mobility ($\mu$) and carrier density ($n$) in the off-state at different temperature were calculated (Fig. 3f), respectively. The mobility in saturated region was extracted by field-effect method (see below). The drift equation $J = qn\mu E$, where $q$ is the elementary charge and $E$ is electric field intensity, is a classical equation to describe the motion of carriers under electric field. I-V characteristics in linear and saturation regime of FET are derived from this equation[28,64,65]. Therefore, the carrier concentration in the off-state ($n_{off}$) can be calculated by off-state current in transfer curves (Fig. 2) or I-V curve of two-terminal measurement (Supplementary Fig. 5) as long as the mobility in the off-state ($\mu_{off}$) is obtained by FET method.

In order to extracted $\mu_{off}$, the dependence of mobility on gate voltage should be considered. DNTT is a thienothiophene small molecule organic semiconductor. Some studies suggest that the organic semiconductors with high degree of delocalization and order, like thienothiophene, show week dependence of mobility on gate voltage[66–69]. The dependence of $\mu$ on ($V_g - V_{th}$) in this work was examined. As shown in Supplementary Fig. 6, the DNTT OEFT exhibits the week dependency of $\mu$ on $V_g$. Therefore, it is reasonable to approximate $\mu$ at $V_g - V_{th} = 0$ as $\mu_{off}$ for estimation of $n_{off}$. Using this approximation, $n_{off}$ was calculated to be about $1 \times 10^{14}\,cm^{-3}$ by drift equation, which is lower than that of organic single crystal[70]. (for details, see Supplementary Note 8). In addition, the carrier concentration of devices in different batches are derived to be in the small range (for details, see Supplementary Table 2). The relative change rate of $\mu_{off}$ and $n_{off}$ with the increasing temperature is shown in Fig. 3f,

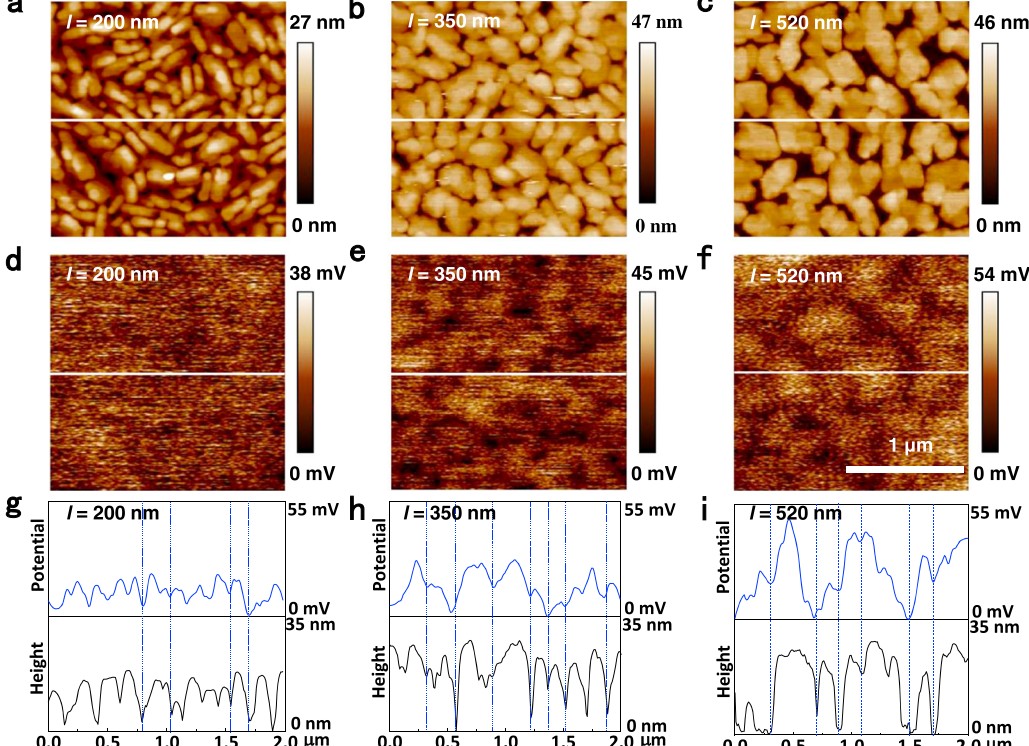

**Fig. 4 The topography and surface potential characteristics of DNTT polycrystalline films measured by Kelvin probe force microscopy (KPFM).**
**a–c** Topography images of DNTT polycrystalline films with grain sizes of 200 nm, 350 nm and 520 nm, respectively. **d–f** Surface potential images of DNTT polycrystalline films. **g–i** Line section profile of topography and surface potential.

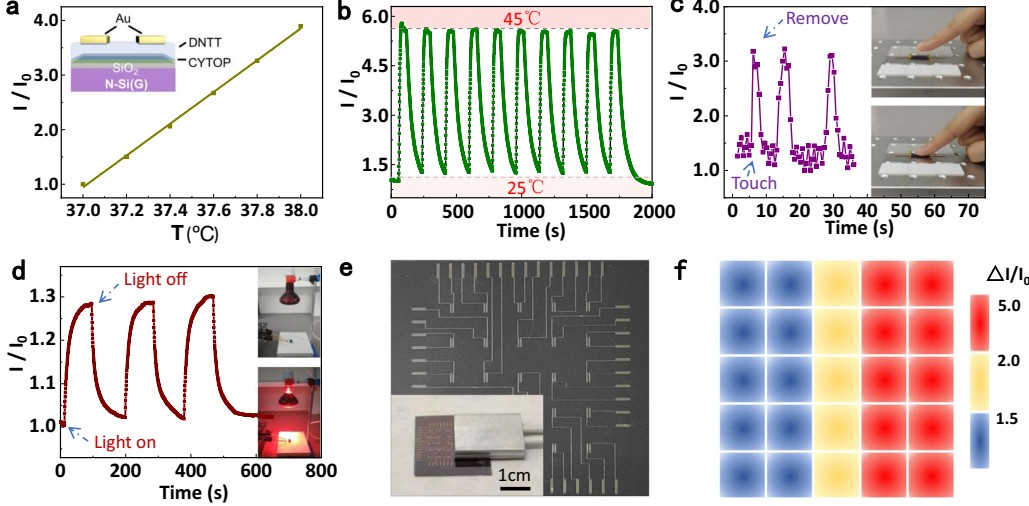

**Fig. 5 Applications of the thermo-sensitive OFET for temperature sensing. a** Temperature resolution curve of sensor measured at 37–38 °C with a precision of 0.2 °C. The inset shows the device structure. **b** Cyclic stability of temperature sensor from 25 °C to 45 °C. **c** Response of temperature sensor to finger touch. **d** Response of temperature sensor to infrared heat source. **e** A photo of the OFET temperature sensor array. **f** Temperature mapping measured by the sensor array.

which conforms to the description of thermionic emission theory. Furthermore, it can be found that $n_{off}$ exhibits larger change, indicating that more carriers obtain enough energy to cross the barriers with the increase of temperature.

From the estimation of $n_{off} \approx 1 \times 10^{14}\,cm^{-3}$ by drift equation (detailed discussion in Supplementary Note 8) and the reported value of DNTT's $\varepsilon_s \approx 3$[71], the twice of Debye length ($2l_D$) was determined to be about 400 nm. It is among the range of DNTT's grain sizes of 200–520 nm in this work. Therefore, the potential barrier ($E_B$) of GBs should be discussed in two conditions. As expressed in Eqs. (2) and (3), when $l < 2l_D$, $E_B \propto l^2$. As the grain size increases to more than $2l_D$ ($l > 2l_D$), $E_B$ becomes independent of $l$. It should be noted that the method of precise calculation of Debye length is not established in organic semiconductors because of the uncertain concentration of dopants. Based on some reasonable assumption and approximation, a way to estimate Debye length is used in this work. It might be not absolutely accurate, but it is in the reasonable range and is enough to qualitatively explain the anomalous temperature dependence in this work. The development of precise calculation methods needs further research in the future.

To further certify the relationship between potential barrier of GB and grain size, the potential of GB should be directly observed. Kelvin probe force microscopy (KPFM) has been proved as a powerful tool for investigating interface electrical structure of organic devices, which can record surface and interface potential distribution of polycrystalline film[72–74]. The potential $\varnothing_B$ of DNTT films with different grain sizes can be obtained directly through KPFM. Figure 4a–c show the AFM images of the DNTT films with grain sizes of 200 nm, 350 nm and 520 nm, respectively, and the corresponding potential distribution images are shown in Fig. 4d–f. The value represents the potential difference between substrate and grains. The curves of height and potential show almost the same profile in these three samples, which directly manifests the congruent relationship between grain sizes and potential (Fig. 4g–i), and confirms that the measured potential is produced from GBs.

The maximum value of potential barrier ($E_B = q\varnothing_B$) is measured to be about 17 meV, 27.8 meV and 47.7 meV for three DNTT films with grain sizes of 200 nm, 350 nm and 520 nm, respectively, which demonstrates the effectiveness of modulating

barrier through grain size. In theory, $E_B$ of 350 nm DNTT film is estimated to be about 9.3 meV by Eq. (2). Although the theorical value of $E_B$ is less than the experimental one, they are in the similar and reasonable range, which can prove the credibility of the theoretical deduction. The analysis of this point was discussed in Supplementary Note 9. Correlating the potential barrier and grain size with the temperature dependence of charge transport of these three DNTT film (Fig. 3a), it can be found that the OFETs with smaller or larger $l$ are relatively thermo-stable, while the OFET with middle $l$ shows great thermo-sensitivity. The reason for this result will be explicitly explained by the effective height of potential barrier in the Discussion Section below.

**Applications of the thermo-sensitive OFET for temperature sensing.** The OFET prepared by thermo-stable semiconductor DNTT has been modulated to be a device with strong thermal dependent property, which is the core performance required by temperature sensors for health monitoring. As a temperature sensor, the OFET shows an high sensitivity of 155 (Fig. 3a, 350 nm). To our knowledge, it is far higher than that of the reported temperature sensors based on OFET[31–35]. In addition, it should be noted that this high sensitivity achieved in the off-state (with current level of about nA) of OFETs produces high input impedance, which usually leads to large noise. This problem could be well-solved by circuit design such as integration of low-noise amplifiers[75]. Benefiting from the high sensitivity, the sensor realizes a quite precise resolution of 0.2 °C with great relative current change from 37 °C to 38 °C (Fig. 5a). It should be pointed out that huge $\triangle I/I_0$ in such small temperature range suggests that our sensor may hold promise for higher resolution. However, due to the limitation of our experimental setup, we could not obtain smaller temperature change in a continuous and reliable mode. Moreover, the cycle stability of the sensor was examined on a semiconductor cooling-heating stage in the temperature range of 25–45 °C. As shown in Fig. 5b, the response signal is stable after multiple cycles. In addition, the sensor also shows quick response and recovery, which is evidenced by two tests. As shown in Supplementary Fig. 7, when finger touched on/off the front side of sensor covered with polyimide film (about 0.06 mm), the response and recover time are 0.58 s and 1.76 s, respectively

(Supplementary Movie 1). When finger touched on/off the back side (about 450 μm silicon), the response and recovery are 2.6 s and 8.2 s, respectively. If without the encapsulation or substrate layer, the sensor may show quicker response and recovery speed.

To demonstrate the ability of our device for some future application such as health care and human-machine interaction, the sensor was used to identify human touch and ambient thermal radiation. The sensor can tolerate the repeated human touch and shows good identification through simple packing with tape (Fig. 5c). Utilizing an IR light as thermal radiation from environment (Supplementary Movie 2), the sensor shows high signal-to-noise ratio and good repeatability (Fig. 5d), which suggests the good capability to sense thermal radiation. In consideration of large-scale integration required by practical application, the temperature sensors were fabricated with a $5 \times 5$ matrix (Fig. 5e) to demonstrate the feasibility for multipoint measurements. The normalized currents of the array were measured when a hot plate was placed on it (inset in Fig. 5e). The mapping of the temperature distribution via the measurement of the normalized drain currents is shown in Fig. 5f, which matches well with the setting position of hot plate. All the application demonstrations indicate the precise temperature sensing capability of the sensor and sensor matrix, suggesting its potential for the practical applications.

## Discussion

To interpret why the OFETs with middle grain sizes show the prominent temperature dependence of charge transport around room temperature, the current density across GB by thermionic emission is investigated theoretically. Through Eq. (6) or definition of differential sensitivity (for details, see Supplementary Note 11), it can be obviously found that the sensitivity at certain temperature shows positive correlation with $(E_F - E_V + E_B - qV_B)$.

As we discussed in Theoretical analysis Section, $E_F - E_V$ makes no difference for the devices with the same materials and processes, so the key parameter influencing sensitivity is the effective height of barrier, $E_{Be} = E_B - qV_B$.

Derived from Eqs. (4) and (5), the Schematic diagram of relationship between $E_{Be}$ and $l$ is shown in Supplementary Fig. 8, which shows two main regimes: (1) $E_B$ generally increases as a quadratic function of $l$ in the case of $l < 2l_D$ (Eq. (4)). (2) When $l > 2l_D$, the traps at GBs are completely filled, rendering $E_B$ to be independent of $l$. However, $V_B$ still enlarges, so $E_{Be}$ begins to decline with $l$ (Eq. (5)). Therefore, maximum of $E_{Be}$ (i.e., sensitivity) should be obtained when $l$ is around $2l_D$, which further causes the strong temperature dependence of charge transport.

In our experiment, $2l_D$ is estimated to be 400 nm, and grain size $l$ varies from 200 nm to 520 nm. Sensitivity increases firstly and then decreases. The maximum sensitivity is obtained when $l$ is 350 nm (Fig. 3a), which is near to $2l_D$ of about 400 nm. These results are consistent with theoretically analysis. To test the universality of this strategy, the OFETs with pentacene as semiconductor layer also implemented the modulation of temperature dependence of charge transport (for method, see Supplementary Note 12). As shown in Supplementary Fig. 9, pentacene film with proper grain size also achieved the larger thermo-sensitivity, demonstrating that the barrier potential regulation is effective to tune thermal activated charge transport of organic semiconductor.

In conclusion, this work has developed an effective strategy to modulate the temperature dependence of thermal activated charge transport in organic semiconductor by precisely tuning the effective height of potential barrier ($E_{Be}$) at GBs (i.e., potential barrier engineering), which has not been reported previously. $E_{Be}$ reaches the maximum when $l \approx 2l_D$, which thus makes the device

highly thermo-sensitive. While refining or coarsening crystalline grain can both reduce $E_{Be}$, which thus enhances the thermo-stability of devices. This result indicates that the grain size should be well optimized into the proper size range according to the thermosensitive or thermostable requirement of applications. The mechanism of modulation for charge transport has been meti-culously elucidated by experimental demonstration and theoretical deduction. The strategy shows good universality for different organic semiconductors. Through the strategy, the DNTT OFET shows an high thermo-sensitivity and can achieve detection of ambient IR, identification of human touch and mapping of temperature distribution. This work brings useful inspiration for both mechanism research of thermal activated charge transport and devices design of integrated circuits and high-performance sensors.

## Methods

**Fabrication of DNTT polycrystalline OFETs**. DNTT was purchased from Sigma-Aldrich and purified three times. CYTOP was purchased from Asahi Glass Cor-poration. The OFETs adopted a bottom-gate top-contact configuration. The highly doped Si wafers (500 μm thick) with 300 nm-thick thermal oxide layer were used as substrate. In order to reduce interface traps, the CYTOP was spin-coated (4000 rpm, 30 s, 180 nm) onto the surface of SiO$_2$ and annealed at 100 °C for 30 min. All DNTT polycrystalline films were vapor deposited on the insulating layer CYTOP surface below $10^{-4}$ Pa. The grain size was controlled by the evaporation rate of DNTT and the substrate temperature, and the thickness was controlled at 20 nm by crystal oscillator. The 260 nm, 350 nm, 420 nm and 520 nm grain sizes were deposited at a rate of about 0.008 nm/s, 0.006 nm/s, 0.004 nm/s and 0.002 nm/s and a substrate temperature of 65 °C, respectively. The 200 nm grain size was deposited at a rate of about 0.01 nm/s and a substrate temperature of 25 °C. The gold-plated electrodes with a thickness of 20 nm were the source-drain electrodes under a vapor deposition rate of about 0.005 nm/s, and the channel length and width were 50 μm and 1000 μm, respectively.

**Characterization of OFETs**. The electrical characterizations of the temperature OFETs were carried out in dark conditions using a Keithley 4200−SCS. In the measurement of transfer characteristic, the drain-source voltage was −60 V, and the gate voltage was scanned from 20 V to −60 V at intervals of 1 V. In finger touch experiment, a finger touched the surface of the sensor which was packaged with tape film. The KPFM measurements were carried out in dark conditions on a Dimension ICON (Bruker), PtIr-coated rectangle Si probe was used. The grain sizes were measured by using the average length of all grains' along the long axes in $2 \times 2$ μm region.

## Data availability

The data presented in this study are available from the corresponding authors on reasonable request.

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

## Acknowledgements

The authors are grateful to National Key Research and Development Program (2018YFA0703200, 2016YFB0401100), National Natural Science Foundation of China (52073210, 21905199, 21573277, 51633006, 62004138), Natural Science Foundation of Tianjin City (19JCZDJC37400, 194214030036), Beijing National Laboratory for Molecular Sciences (BNLMS202006) and Key Research Program of Frontier Sciences of Chinese Academy of Sciences (QYZDB−SSW−SLH031). The authors highly appreciate Prof. Jun Kang (Beijing Computational Science Research Center), Dr. Wenchong Wang (University of Munster), Prof. Xiaojun Guo (Shanghai Jiao Tong University), Prof. Chengliang Wang (Huazhong University of Science and Technology), Prof. Tao Li (Shanghai Jiao Tong University), Prof. Yanyan Fu (Shanghai Institute of Microsystem and Information, Chinese Academy of Sciences), who provided valuable and constructive comments and suggestions for the theoretical analysis in this work.

## Author contributions

X.G. and Y.H. performed the sensor fabrication and measurements. Y.M., Z.W., X.C., J.L. and D.J. assisted in the experiments. Z.W. and W.H. made valuable suggestions on this work. L.L., Y.H. and X.G. conceived, analyzed and wrote the paper. L.L. conceived and supervised this work.

## Competing interests

The authors declare no competing interests.
