## [Peer Review File · Nature Communications]

Reviewers' Comments:

Reviewer #1:

Remarks to the Author:

The authors presented an interesting temperature dependence of charge transport in a DNNT, and demonstrated extremely sensitive temperature sensors. The strategy proposed by the authors is to precisely tune the barrier potential of grain boundaries to be thermal energy of room temperature. At first glance, overall results seem to be interesting in terms of device physics and material science, and are worthy publication in Nature Communications.

However, I cannot judge whether the authors' analysis and interpretation associated with the temperature dependent charge transport are correct or not, because of poor presentation of the manuscript in my hands. I am afraid to say that most information necessary to justify the result is fatally missing in the manuscript, and the manuscript in my hands never reaches the level required by a standard scientific paper.

I would not accept this work at this stage, and kindly ask the authors to clarify details below;

Figures

1. In Figs. 1, 2, and 3, how much is the drain voltage applied to OFETs? In order to estimate the conductivity and mobility accurately, it should be very important to identify whether the present OFET is in a saturation regime or linear regime during the analysis.
2. In Fig.3, the expression of sensitivity, $\Delta I/I_0$, is really misleading. This should be modified to $[I(V_g, T) - I(V_g, T_0)]/I(V_g, T_0)$, where T_0 is the reference (base) temperature (is this correct?). It is because the sensitivity is gate-voltage, and temperature variant. In addition, the author should specify the values of V_g , T_0 , and T in Figs.3a and c.
3. The definition of sensitivity should be address in the different way; we often use the temperature coefficient of resistance (TCR in the unit of degC⁻¹). The value for the best DNNT device with the grain boundary of 350 nm is likely to show remarkably high TCR (approx. 2.5 from my calculation). The authors should clarify how this value is comparable to those obtained for conventional temperature sensors.
4. In Fig. 3c, what is the temperature difference for this plot (presumably 358 - 298 = 60 K)?
5. In Figs. 3d and e, these plots are misleading. First of all, from d the T-dependence of I_{ds} looks $I_{ds} \exp(T)$, whereas from e I_{ds} follows the activation behavior. The authors should double-check the validity of fittings here, and clarify the resulting fitting parameter in Supplementary information. Obviously, the most important parameter, the height of potential barrier at grain boundary EB, can be determined from the fitting. The author should clarify all the values EB for the DNNT films having 200, 350, and 520 nm, and compare these values with those obtained from SKPM measurements. By the way, in Fig.3e is a vertical axis $\ln(I)$ or $\ln(\sigma_{off})$? The latter is written in the main tex. Related to this, I am skeptical about the estimation of conductivity σ_{off} . Presumably, σ_{off} was derived from I_{off} being divided by V_{ds} . If so, this may lead an incorrect estimation because the equation $\sigma_{off} = I_{off} / V_{ds}$ assumes that the potential drop follows linearly along the channel direction. In the off-state of OFETs, it should be more reasonable to employ equation (1) or simply the expression of SCLC.
6. In Fig. 3f, I do understand that the message of this panel is that the excellent temperature sensitivity present in the manuscript originates from the large change in the carrier density n . However, I am skeptical about estimations of mobility μ and charge density n . How did the author estimate the mobility at such low doping level? If the charge transport actually follows the thermionic emission model, the mobility should be field-dependent, meaning that the field-effect mobility derived from transfer characteristics is no longer applicable. In addition, at this low charge density, $\sigma_{off} = ne\mu$ is nontrivial.

I agree with this argument because at such low doping levels n changes sensitively with respect to temperature; an increase of kBT effectively changes the Fermi energy relative to the transport level. The author should double check how much the Fermi level shift can be expected when a simple exponential density-of-states is assumed.

7. The estimation of the Debye screening length is incorrect. The Debye length derived by the authors is unrealistically large. It is normally on the order of 10 nm for organic semiconductors. Presumably, the authors made a fatal mistake on conversion from cm to m, or conversion from 2D to 3D charge density. From my rough estimation with n of $10^{18}\sim 10^{19}$ cm⁻³ (the typical value of n that is induced at the OFET devices), the Debye length can be a few nm. This gives a conceptual failure because the grain boundary engineering never works on the precise tune of barrier height. The author should double-check the values.

8. In equations (4)-(5), the author should explain more about the assumptions used here. In a semiconductor textbook, $n(V_g, T) = N_c(T) \exp(-E_B/kBT)$ is often found for instance a textbook written by Sze, where $N_c(T)$ is temperature variant. Although I am not confident about how this expression is applicable to organic semiconductors, particularly in the thermionic emission model, the authors should take more care of taking the derivative with T (i.e., dn/dT). It will be appreciated that the authors will communicate with semiconductor physicist to clarify why 26 meV (thermal energy of RT) plays a vital job on maximizing sensitivity.

9. There are minor typos and errors found throughout the manuscript; for example, - exp (exponential) is found to be e^{px} ,
- °C is missing in Fig.5

Overall, mainly due to unrealistic estimation of the Debye length, I hardly believe that the proposed grain boundary engineering can tune the potential height. On the other hand, SKPM measurements showed a reasonable trend that larger GB will give larger potential barrier. Again because of lack of important information, I would not be able to judge whether the authors' analysis and interpretation associated with the temperature dependent charge transport are correct or not. After the authors address all the comments above, I will be happy to review the manuscript again.

Reviewer #2:

Remarks to the Author:

Their manuscript entitled "Effectively Modulating Thermal Activated Charge Transport in Organic Semiconductors by Precise Potential Barrier Engineering" is focusing on studying the grain size effect on the transfer properties of the DNTT transistors and utilize the grains with different sizes and boundary density to achieve temperature sensor functions. The overall quality of the work is good and it should be published in nature communication. It covers from fundamental science to practical applications. Using the grain size effect to achieve temperature sensing is a very good idea. Here I summarized some of the comments for the authors to further improve the manuscript:

1. Figure 3(d) is I_{off} , and it maybe confusing if the axis title used I_{DS} .
2. The extraction of the mobility and carrier concentration in figure 3f is a bit confusing to me. Did the author use mobility and conductivity to get the carrier concentration or did they use the carrier concentration and conductivity to get the mobility? If it is the first case, how did they calculate the mobility? If it is the 2nd case, how did they calculate the carrier concentration?
3. The structure of the device used in figure 5 should be given. Does it have the same structure as the device used in Figure 3c? What gate bias is used to measure the ΔI ?
4. More information about the origin of equation (2) and (3) would be useful.

Reviewer #3:

Notes: Figure * denotes the Figure in the main text, Figure R* denotes Figure in the reply for reviewer below, and Figure S* denotes Figure in the Supplementary Information. For clarity and convenience, several Figures appear twice in our reply for different comments

Reply for Reviewer #1

Comments 1: The authors presented an interesting temperature dependence of charge transport in a DNNT, and demonstrated extremely sensitive temperature sensors. The strategy proposed by the authors is to precisely tune the barrier potential of grain boundaries to be thermal energy of room temperature. At first glance, overall results seem to be interesting in terms of device physics and material science, and are worthy publication in Nature Communications. However, I cannot judge whether the authors' analysis and interpretation associated with the temperature dependent charge transport are correct or not, because of poor presentation of the manuscript in my hands. I am afraid to say that most information necessary to justify the result is fatally missing in the manuscript, and the manuscript in my hands never reaches the level required by a standard scientific paper. I would not accept this work at this stage, and kindly ask the authors to clarify details below.

Our reply: Thank you very much for your high and constructive comments and suggestions, which guide us to think some points deeply, improve manuscript, correct some errors, and make some unclear points clear. After getting your comments, we have checked lots of relevant literatures and books, and also discussed with six experts in semiconductor physics and organic semiconductors (listed at the end of our reply).

For clarity and convenience, the key explanations and revisions are listed as follows:

- (1) According to thermionic emission theory, the current density equation and fitting curve were modified. (**Our reply for comment 6**).
- (2) The dependence of mobility on gate voltage and drain-source voltage was

discussed. (**Our reply for comment 7**).

(3) The estimation of carrier density and Debye length was discussed in detail. (**Our reply for comment 7 and 8**).

(4) The reason why devices with moderate grain size show maximum sensitivity (i.e., temperature dependence) was reinterpreted by the deeper understanding and analysis of thermionic emission theory. (**Our reply for comment 9**).

(5) Temperature coefficient of resistance (TCR) of the OFETs was calculated and discussed.

(6) Some errors and unclear points were modified, and some data were added.

Below we would like to address your comments one by one.

Comments 2: In Figs. 1, 2, and 3, how much is the drain voltage applied to OFETs? In order to estimate the conductivity and mobility accurately, it should be very important to identify whether the present OFET is in a saturation regime or linear regime during the analysis.

Our reply: Thanks for your meticulously observation! The omission of the drain voltage in manuscript make some important parameters unclear. We point out that all the drain voltages applied to OFETs in transfer characteristic measurement are -60 V and the present OFET is in the saturation regime.

Our revisions: We have added the information into main text and relevant Figures

Comments 3: In Fig. 3, the expression of sensitivity, $\Delta I/I_0$, is really misleading. This should be modified to $[I(V_g, T) - I(V_g, T_0)]/I(V_g, T_0)$, where T_0 is the reference (base) temperature (is this correct?). It is because the sensitivity is gate-voltage, and temperature variant. In addition, the author should specify the values of V_g , T_0 , and T in Figs.3a and c.

Our reply: Thanks for the kind reminder! According to your suggestion, we have

modified the misleading expression of sensitivity to $[I(V_g, T) - I(V_g, T_0)]/I(V_g, T_0)$, where T_0 is the base temperature. The value of V_g is 20 V in Fig. 3a. The values of T_0 , and T are both 298K and 358K in Fig. 3a and c.

In addition, it should be noted that the sensitivity calculated in the off-state regime (i.e., V_g in the range of 0~20 V) is at the similar level since the off-state current of OFET exhibits flat and smooth curves. For example, the sensitivity at $V_g = 0$ V (**Fig. R1** i.e., Fig. 3c in main text) is quite similar to that at $V_g = 20$ V.

Our revisions: We have modified the misleading expression of sensitivity to $[I(V_g, T) - I(V_g, T_0)]/I(V_g, T_0)$, where T_0 is the base temperature. The value of V_g is 20 V in Fig. 3a. The values of T_0 , and T are both 298K and 358K in Fig. 3a and c. In addition, the similar sensitivity at the off-state regime is discussed briefly in the main text, and the corresponding detailed discussion is added in the Supplementary Information.

Fig. R1. Sensitivity (298 K-358 K, i.e., ΔT is 60K) varies with the gate voltage of the OFET.

Comments 4: The definition of sensitivity should be address in the different way; we often use the temperature coefficient of resistance (TCR in the unit of degC^{-1}). The value for the best DNTT device with the grain boundary of 350 nm is likely to show remarkably high TCR (approx. 2.5 from my calculation). The authors should clarify how this value is comparable to those obtained for conventional temperature sensors.

Our reply: Thanks for your valuable and kind suggestions! The temperature coefficient of resistance (TCR) is usually defined as follows:

$$\text{TCR} = \frac{1}{R} \frac{dR}{dT} \times 100\% \quad (1)$$

where R is the base resistance at 298K. The TCR curves of the DNTT devices with different grain sizes are shown in **Fig. R2**. R decreases exponentially with increase of T due to thermal activated charge transport, so TCR value also decreases with T¹. However, the values are higher than, at least comparable to, most conventional temperature sensors. For example, TCR for most metals is between 0.1–1% K⁻¹₂, and similarly, other conductive materials, such as single-walled carbon nanotubes (SWCNTs)³ and poly(3,4-ethylenedioxythiophene) polystyrene sulfonate (PEDOT:PSS)⁴, also exhibit a comparable sensitivity range. Most oxide ceramics shows the value of TCR 2-4% K⁻¹_{5,6}

Fig. R2. The TCR of the DNTT devices with the grain size of 200 nm, 260nm, 350 nm, 420nm and 520 nm.

Furthermore, we also calculated the average value of TCR by:

$$\text{TCR}_{avg} = \frac{1}{R_0} \frac{R - R_0}{T - T_0} \times 100\% = \left(\frac{R}{R_0} - 1 \right) / \Delta T \times 100\% \quad (2)$$

where R_0 and R are the value of resistance at 298K and 358K, respectively, and ΔT is

60 K. The TCR_{avg} of the DNTT devices with the grain size of 200 nm, 260 nm, 350 nm, 420 nm and 520 nm are -1.38%, -1.65%, -1.66%, -1.65%, and -1.59%, respectively. From this data, it can be found that the TCR value of devices with different grain size exhibits tiny distinction, but this fact does not indicate that these devices show similar sensing performance. The tiny distinction in TCR value mainly stems from the calculation method, which can be understood as follows:

In fact, if the resistance change is small, the TCR is suitable for reflecting the sensing performance. On the other hand, if the resistance change is big (e.g. over one order of magnitude) and negative, TCR would not be the best choice to characterize the sensitivity.

We can take a simple example to explain. Assuming two sensors have the same base resistance (R_0) of 100 Ω (which produces a current (I_0) of 1A at voltage of 100 V). In the first sensor the resistance (R) changes to 10 Ω (producing a current (I) of 10 A at voltage of 100 V) when temperature increases by 10 $^{\circ}\text{C}$. In the second sensor R changes to 1 Ω (100 A at voltage of 100 V) at the same temperature change of 10 $^{\circ}\text{C}$. Apparently, the second sensor exhibits significantly better sensing performance than the first one. The TCR value of these two sensors is calculated to be $-9\% \text{ }^{\circ}\text{C}^{-1}$ and the $-9.9\% \text{ }^{\circ}\text{C}^{-1}$, respectively. Unfortunately, TCR value does not clearly reflect the big difference in sensing performance between these two sensors. If the sensitivity is defined by the relative current change, $((I - I_0)/I_0)$, these two sensors will exhibit sensitivity of 9 and 99, respectively, which can clearly show the big difference in sensing performance.

In our work, the output signal is current, and increases over one of magnitude when the temperature increases by tens of degree. Therefore, in the initial manuscript, we adopt relative current change, $[I(V_g, T) - I(V_g, T_0)]/I(V_g, T_0)$, as sensitivity (ΔT is 60 $^{\circ}\text{C}$). Undoubtedly, as you stated, TCR as the standard sensitivity parameter is highly useful for making comparison with other sensors. Therefore, we use both sensitivity in the revised manuscript. **Table R1** lists the sensitivity value defined via TCR and

our method.

Table R1. Sensitivity value defined via TCR and our method

Grain size	200nm	260nm	350nm	420nm	520nm
$TCR_{avg} (^{\circ}C^{-1})$	-1.38%	-1.65%	-1.66%	-1.65%	-1.59%
$\Delta I/I_0$	5	90	155	80	21

Our revisions: We have calculated the value of TCR of the DNTT devices with different grain size. The corresponding data and discussion are added into the main text and Supplementary Information.

Comments 5: In Fig. 3c, what is the temperature difference for this plot (presumably $358 - 298 = 60$ K)?

Our reply: Thanks for pointing out this unclear point! The temperature difference in Fig. 3c is 60 K (298 K-358 K).

Our revisions: We have marked the temperature difference in relevant Figures.

Comments 6: In Figs. 3d and e, these plots are misleading. First of all, from d the T -dependence of I_{ds} looks $I_{ds} \propto \exp(T)$, whereas from e I_{ds} follows the activation behavior. The authors should double-check the validity of fittings here, and clarify the resulting fitting parameter in Supplementary information. Obviously, the most important parameter, the height of potential barrier at grain boundary E_B , can be determined from the fitting. The author should clarify all the values E_B for the DNTT films having 200, 350, and 520 nm, and compare these values with those obtained from SKPM measurements. By the way, in Fig.3e is a vertical axis $\ln(I)$ or $\ln(\sigma_{off})$? The latter is written in the main tex. Related to this, I am skeptical about the estimation of conductivity σ_{off} . Presumably, σ_{off} was derived from I_{off} being divided by V_{ds} . If so, this may lead an incorrect estimation because the equation $\sigma_{off} = I_{off} / V_{ds}$

assumes that the potential drop follows linearly along the channel direction. In the off-state of OFETs, it should be more reasonable to employ equation (1) or simply the expression of SCLC.

Our reply: Thanks for your constructive comments which enable us to inspect experimental analysis, deepen the understanding, and correct some errors. These comments mainly concern two aspects: (1) the fitting between current (I) and temperature (T), and (2) the estimation of conductivity σ_{off} . We would like to address these two aspects as follows:

(1) **The fitting method:**

We re-checked the thermionic emission theory, did the proper analysis and transformation of the equation, and found a proper fitting method for our results. Below are the details:

According to Bethe's thermionic emission theory^{7,8}, the net current in Schottky barrier is a superposition of two currents. One is the current from semiconductor to metal, and the other is the current from metal to semiconductor. If the applied voltage is forward bias (**Fig. R3**), the current density from semiconductor to metal can be given by the following equation:

$$J_{n \rightarrow m} = A^* T^2 \exp\left(-\frac{q\phi_{Bn} - qV}{k_B T}\right) \quad (3)$$

where A^* is effective Richardson constant, and $q\phi_{Bn}$ is Schottky barrier. Because the height of barrier for electrons from metal to semiconductor is independent on bias voltage, the current density from metal to semiconductor is

$$J_{m \rightarrow n} = -A^* T^2 \exp\left(-\frac{q\phi_{Bn}}{k_B T}\right) \quad (4)$$

The total current density is the sum of Eq. (3) and (4)

$$\begin{aligned} J &= J_{n \rightarrow m} + J_{m \rightarrow n} \\ &= A^* T^2 \exp\left(-\frac{q\phi_{Bn} - qV}{k_B T}\right) - A^* T^2 \exp\left(-\frac{q\phi_{Bn}}{k_B T}\right) \end{aligned}$$

$$= A^*T^2 \exp\left(-\frac{q\phi_{Bn}}{k_B T}\right) \left[\exp\left(\frac{qV}{k_B T}\right) - 1 \right] \quad (5)$$

Fig. R3. The thermionic emission theory of Schottky barrier based on metal-semiconductor contact. (Sze, S. M. Physics of semiconductor devices. Wiley 2006)

The Schottky barrier $q\phi_{Bn}$ is defined as the energy difference between the conduction band and the Fermi level, i.e., $q\phi_{Bn} = E_C - E_{Fn}$ (**Fig. R3**). In our model of potential barrier at grain boundary (**Fig. R4**), E_B is the height of the potential barrier at GBs, which is equal to the degree of band bending. Then $q\phi_{Bn}$ is the same meanings as $E_F - E_V + E_B$ (E_V is HOMO without bending) in our model. Therefore, the current density across a grain boundary should be given by

$$J = A^*T^2 \exp\left(-\frac{E_F - E_V + E_B}{k_B T}\right) \left[\exp\left(\frac{qV_B}{k_B T}\right) - 1 \right] \quad (6)$$

where V_B is the voltage drop across a GB. The contribution of minority carrier (i.e., electron) is negligible and all discussions about carriers is for majority carrier (i.e., hole) in this paper.

Furthermore, the height of potential barrier on both sides of GBs is changed by bias voltage⁸. As shown in **Fig. R4**, the effective height of barrier (mentioned as E_{Be}) on the forward bias side is $E_B - qV_B$ and that on the reverse bias is $E_B + qV_B$. Therefore, charge carriers on the reverse bias can hardly cross the higher potential barrier, and the corresponding current is negligible. Eq. (6) can be approximated to be the following Eq. (7).

$$J = A^*T^2 \exp\left(-\frac{E_F - E_V + E_B - qV_B}{k_B T}\right) \quad (7)$$

Eq. (7) can also be derived by the approach of Rhoderick and Williams (Metal-semiconductor contacts, 2nd Ed. Clarendon, Oxford **1988**)⁹.

Fig. R4. Schematic diagrams of energy band model at grain boundary of polycrystalline semiconductor with a bias.

The resistance of a polycrystalline material consists of the contributions from the grain-boundary region (R_B) and the bulk of the crystallite (R_C). We assume that the DNTT film is composed of identical crystallites with a grain size of l . The grains are also assumed to be seamless connection. The channel (with length of L) contains L/l grains and GBs, and then V_B can be described as

$$V_B = V_{ds} \frac{R_B}{R_B + R_C} \frac{l}{L}, \quad 0 < l \ll L \quad (8)$$

where V_{ds} is the voltage drop of the channel. If we define $\beta = \frac{R_B}{R_B + R_C}$, Eq. (8) becomes

$$V_B = \beta V_{ds} \frac{l}{L}, \quad 0 < l \ll L, \quad 0 < \beta < 1 \quad (9)$$

Dividing both sides by T^2 and taking the logarithm, Eq. (7) becomes

$$\ln \frac{J}{T^2} = \ln A^* - \left(\frac{E_F - E_V + E_B - qV_B}{k_B} \right) \frac{1}{T} \quad (10)$$

Eq. (10) suggests $\ln \frac{J}{T^2} \propto \frac{1}{T}$. Then, we draw the plot of $\ln \frac{J}{T^2}$ as a function of $1000/T$ for the device with grain size of 350 nm in **Fig. R5**. It shows a good fit with Eq. (10), suggesting that the charge transport in the DNTT film accords with the thermionic emission model. The critical parameter $\frac{E_F - E_C + E_B - qV_B}{k_B}$ can be obtained from the slopes of these fitting curves, but E_B can not be directly calculated because of unclear value of β and $E_F - E_C$. The fitting curves of 200 nm and 520 nm DNTT film are added into Supplementary Fig. 4, both of which shows good fit with the thermionic emission model. In addition, these results also show that steeper slope indicates higher temperature sensitivity (i.e., 350 nm > 520 nm > 200 nm). The relation of temperature dependence and fitting slope (i.e., $\frac{E_F - E_C + E_B - qV_B}{k_B}$) will be discussed in detail at Our reply for Comments 9.

Fig. R5. The plot of $\ln \frac{J}{T^2}$ as a function of $1000/T$.

(2) Estimation of conductivity:

We reconsidered the estimation of conductivity σ_{off} according to your suggestions. The current-voltage characteristics do not accord with ohmic law in the off-state, so calculation of conductivity by $\sigma_{\text{off}} = I_{\text{off}} / V_{ds}$ might be questionable. In fact, the estimation of conductivity is used to calculate the carrier density n in off-state. To avoid the questionable calculation of conductivity, we find another reliable method to

calculate n by drift equation. It will be discussed at Our reply for Comments 7. Therefore, we will no longer calculate the conductivity.

Our revisions: We have modified Eq. (1) in main text, and replaced the incorrect fitting data. The discussion about fitting is added into main text. The estimation of conductivity is deleted.

Comments 7: In Fig. 3f, I do understand that the message of this panel is that the excellent temperature sensitivity present in the manuscript originates from the large change in the carrier density n . However, I am skeptical about estimations of mobility μ and charge density n . How did the author estimate the mobility at such low doping level? If the charge transport actually follows the thermionic emission model, the mobility should be field-dependent, meaning that the field-effect mobility derived from transfer characteristics is no longer applicable. In addition, at this low charge density, $\sigma_{off} = ne\mu$ is nontrivial. I agree with this argument because at such low doping levels n changes sensitively with respect to temperature; an increase of $k_B T$ effectively changes the Fermi energy relative to the transport level. The author should double check how much the Fermi level shift can be expected when a simple exponential density-of-states is assumed.

Our reply: Thanks for your professional comments, which together with other comments help us to find a new and rational way to explain our results! Your comments mainly include three aspects: (1) estimations of mobility μ ; (2) estimation of carrier density; (3) Fermi level shift.

(1) Estimations of mobility μ

We firstly address your doubts about the estimations of mobility μ . We used the field-effect method to calculate the saturated region mobility by

$$I_{ds} = \frac{W}{2L} \mu_{sat} C_i (V_g - V_{th})^2 \quad (11)$$

As you stated, the mobility in saturation regime (μ_{sat}) is indeed not equal to the

mobility μ in the off-state with very low charge density n , because μ often has been reported to depend on n in organic semiconductors. However, DNTT is a thienothiophene small molecule organic semiconductor. These molecules are highly delocalized, closely stacked and a low degree of disorder^{10,11}. The researches by Liu suggest that the $\mu - n$ relation in OFETs is closely related to the degree of delocalization and ordering degree of organic semiconductors^{12,13}. **Fig. R6** shows the correlation between the dependency of μ on $V_g - V_{th}$ (corresponding to the dependency of μ on n) and the degree of delocalization in organic semiconductors. We also extracted the $\mu - (V_g - V_{th})$ relation from transfer characteristic of the DNTT device as shown in **Fig. R7**, which indicates the week dependency of μ and V_g . The μ decreases by less than an order of magnitude from on-state ($V_g - V_{th} = 20V$) to off-state ($V_g - V_{th} = 0$). It is similar to TIPS-pentacene with $\Delta D=0.7$ and $\square=0.36$ (red line in **Fig. R6 a** and purple triangle in **Fig. R6 a**). We used the μ at $V_g - V_{th} = 0$ (the change of μ is negligible when $V_g < V_{th}$) as the estimated value of μ in the off-state to approximately calculate charge density n , which would not cause a large deviation in the estimation of n .

Fig. R6. a) The calculated mobility as a function of gate voltage for different ΔD . b) The values of ΔD and ΔE are plotted for different semiconductors. The gray arrow shows the direction of increasing disorder in charge transport. (Chuan Liu et. al *Mater. Horiz.* **2017**, 4, 608)

Fig. R7. The dependence of μ on $(V_g - V_{th})$ of the DNTT OFET.

The above discussions indicate that the field-effect mobility shows dependence on the gate voltage (vertical electric field), and that a method to estimate the mobility in off-state is deduced according to the literatures. Furthermore, we totally agree with your arguments that mobility should be field-dependent. In thermionic emission model, the mobility is supposed to be dependent on the drain-source field (lateral electric field) as the following equation¹⁴,

$$\mu_{eff} = \mu_0 \exp(\gamma \sqrt{E}) \quad (12)$$

where μ_0 and μ_{eff} are intrinsic mobility and effective mobility, and $E = \frac{V_{ds}}{L}$ is average electric field intensity in the channel. The mobility measured by FET characterizations is μ_{eff} . The channel length of all the devices is 50 μm and the applied V_{ds} is always -60 V, so the field-dependent of mobility in our experiment is not reflected.

In fact, there are several methods to characterize the mobility of organic semiconductors such as FET, SCLC, and Hall. However, it should be noted that FET characterization is the most reasonable way we can take to estimate μ in the off-state. Below we will explain why it is highly difficult to measure proper mobility with

SCLC and Hall in this situation:

①. SCLC is a common way to test mobility, but it encounters difficulties to measure the mobility along the π -stacking direction of organic field-effect semiconductors, which can be understood as follows:

Organic field-effect semiconductors have anisotropic charge transporting property. Generally, the mobility along the π -stacking direction is greatly higher than that perpendicular π -stacking direction (**Fig. R8**). Sirringhaus et al.¹⁵ found the mobility of the same polymer varies by four orders of magnitude in different molecular orientations (**Fig. R9**). In our previous work¹⁶, we also reported that phthalocyanine molecules with edge-on orientation shows significantly higher mobility than that of face-on orientation (**Fig. R10**). In addition, most of organic field-effect semiconductor adopt edge-on molecular orientation, i.e., the π -stacking direction is parallel with the substrate and consistent with the charge transporting direction. The edge-on feature is a key for obtaining high mobility of OFET. In our devices, XRD measurements confirm that DNTT also adopt edge-on orientation, which is beneficial for charge transporting. Therefore, if we want to measure the mobility along π -stacking direction of DNTT with other methods, it is necessary to guarantee that the charge transporting direction in the measurement is consistent with π -stacking direction.

Fig. R8. The schematic diagram of (a) edge-on and (b) face-on molecular orientation

Fig. R9. The mobility varies with molecular orientations. (Sirringhaus et al., *Nature* **1999**, 401, 685-688)

Substrate	Substrate Temperature[°C]	Phase State	Molecular orientation	Morphology	Mobility[cm ² V ⁻¹ s ⁻¹]
SiO ₂	RT	amorphous	Random	Continuous Small grain	10 ⁻⁴
SiO ₂	120	Crystalline	Face-on	Discontinuous cone array	10 ⁻⁶
OTS	120	Crystalline	Edge-on	Continuous Large grain	0.06

Fig. R10. The schematic diagram and mobilities of Face-on and Edge-on molecules.

(Li, L et al., *Adv. Energy Mater.* **2011**, 1, 188-193)

SCLC method usually uses a sandwich's structure device¹⁷ (**Fig. R11**), whose charge transport direction is perpendicular to organic semiconductor film. As stated above, the organic field-effect semiconductors always adopt edge-on molecular orientation. If SCLC measurement is performed on organic field-effect semiconductors film with edge-on orientation, it may measure the charge transporting mobility perpendicular to the π -stacking direction, which would be several orders of magnitude lower than that along the π -stacking direction. This value would not be useful for our current work. If we want to measure mobility along the π -stacking direction with SCLC method, we need to prepare DNTT film with face-on molecular orientation, which is highly difficult or even impossible task because only few organic field-effect semiconductors may adopt face-on orientation under special growth conditions. Until now, all the reported DNNT film adopt edge-on orientation^{18,19,20,21}.

Fig. R11. The device structure of SCLC. DNTT normally forms edge-on molecule orientation. Charge transport is vertical to π -stacking direction in the SCLC structure.

② Hall effect measurement is normally suitable for semiconductors with high mobility; otherwise the Hall voltage might be overshadowed by the noise such as thermoelectric voltage and misalignment voltage. Therefore, in organic semiconductors, Hall measurement is always performed on single crystal system with mobility of several to tens of cm^2/Vs ^{22,23,24}. Since DNTT polycrystalline films in our work exhibit mobility about $1 \text{ cm}^2/\text{Vs}$, it would be highly difficult to measure the Hall voltage and further calculate mobility. In addition, Hall mobility describes the intrinsic transport of free charge carries, which does not match the DNTT devices in

this work.

Above all, field-effect method is still reasonable way to measure the mobility of DNTT film in the off-state regime. Furthermore, DNTT film is highly ordered (confirmed by XRD measurements), so the mobility shows weak dependence on the voltage, which has been demonstrated by the literatures¹⁰⁻¹³. Therefore, we use the mobility at $V_g - V_{th} = 0$ to calculate other parameters in the off-state regime.

(2) Estimation of carrier density

We agree with your comment about the equation ($\sigma_{off} = ne\mu$). This equation is derived from generalized ohm's law, and the $I-V$ relation in the off-state don't obey ohm's law. In order to estimate charge density n , we use the drift current equation. The current density is defined as the amount of charge that passes through a unit area per unit time.

$$J = \frac{Q}{St} \quad (13)$$

where Q is quantity of charge, and S is sectional area. Assuming that the length of charge drifting in time t is L , Eq. (13) becomes

$$J = \frac{QL}{SLt} = \frac{QL}{Vt} \quad (14)$$

Inserting $\frac{Q}{V} = qn$ and $\frac{L}{t} = v$ into Eq. (14), it becomes

$$J = qnv \quad (15)$$

Where n is volume density of charge carrier, and v is drift velocity of charge carrier. Using $v = \mu_{eff}E$ with Eq. (15), the current density can be given by

$$J = qn\mu_{eff}E \quad (16)$$

The carrier density in off-state can be estimated through Eq. (16) with field-effect mobility extracted by transfer curve in off-state ($V_g - V_{th} = 0$). We have recalculated the n_{off} of the DNTT OFETs and modified Fig. 3f.

(3) Fermi level shift

According to your comments, we estimated the Fermi level shift in the depleted region near GBs by Eq. (17)⁸

$$E_F - E_V + E_B \approx kT \ln \left(\frac{N_V}{n} \right) \quad (17)$$

where $N_V = 2 \left(\frac{2\pi m_p^* k_B T}{h^2} \right)^{\frac{3}{2}} = 2 \left(\frac{2\pi m_0 k_B T}{h^2} \right)^{\frac{3}{2}} \left(\frac{m_p^*}{m_0} \right)^{\frac{3}{2}}$. m_0 is the electron rest mass, and m_p^* is the hole effective mass. m_p^*/m_0 is set to be 2.5 for DNTT according to the previous report²⁵. Using n of the DNTT OFETs at a temperature range of 298-358K, the Fermi level shift ΔE_F was estimated to be -45 meV, i.e., E_F shifted 45 meV to the valence band.

Our revisions: We modified the estimation of mobility and carrier density in the off-state, and estimated the Fermi level shift. The corresponding data and discussion were added into main text and Supplementary Information.

Comments 8: The estimation of the Debye screening length is incorrect. The Debye length derived by the authors is unrealistically large. It is normally on the order of 10 nm for organic semiconductors. Presumably, the authors made a fatal mistake on conversion from cm to m, or conversion from 2D to 3D charge density. From my rough estimation with n of $10^{18} \sim 10^{19} \text{ cm}^{-3}$ (the typical value of n that is induced at the OFET devices), the Debye length can be a few nm. This gives a conceptual failure because the grain boundary engineering never works on the precise tune of barrier height. The author should double-check the values.

Our reply: Thanks for your comments on the Debye length, which enable us to think it deeply. Below we would like to explain it in details.

Debye length is defined as

$$l_D = \sqrt{\varepsilon_s \varepsilon_0 k_B T / q^2 n} \quad (18)$$

It is noticed that volume charge density n is a critical paramant in question. The n

of $10^{18}\sim 10^{19} \text{ cm}^{-3}$ is indeed reported at the OFET devices. However, it is the typical value for deliberately doped organic semiconductors^{26,27} or the OFETs in the on-state (V_g induced a mass of charge carriers)^{28,29}. The areal carrier density in on-state can be given by $n_{on} = \frac{1}{q} C_i (V_g - V_{th})$ ¹³, where C_i is capacitance per unit area of dielectric. The C_i of common dielectric materials is a few or tens nF/cm² and $V_g - V_{th}$ varies from several to tens of V. As a result, n_{on} is about $10^{11}\sim 10^{13} \text{ cm}^{-2}$, i.e., volume density is about $10^{17}\sim 10^{19} \text{ cm}^{-3}$ assuming the uniform distribution of carriers in conducting channel.

In our experiment, the discussion of temperature dependence is in the off-state, so the charge density n_{off} is much lower than $10^{18}\sim 10^{19} \text{ cm}^{-3}$. According to Eq. (16), the n_{off} of the DNTT devices are estimated to be $10^{13}\sim 10^{14} \text{ cm}^{-3}$, which is reasonable and also accords with the previous reports^{30,31}. In addition, this density range can be further verified by the following considerations: Generally, the on/off rate of OFETs is about 10^5 in our work. The increased conductance is mainly contributed from carriers induced by high V_g , considering weak dependence of mobility on V_g ($\mu_{on}/\mu_{off} < 10$). Assuming the n_{on} of $10^{18}\sim 10^{19} \text{ cm}^{-3}$, the value of n_{off} can be roughly estimated to be $10^{13}\sim 10^{14} \text{ cm}^{-3}$.

It should be pointed that the n_{off} here is density of free carriers, and the n for calculating Debye length should be density of total carriers^{32,33,34}. The carriers in off-state is mainly derived from impurities that act as dopants. In our experiment, the DNTT OFETs were fabricated in the similar condition (only evaporation rate and substrate temperature were tuned), so the doping density should be consistent. Nevertheless, the estimation of n_{off} shows decreasing trend with the increase of grain size, which may be explained by the fact that larger barrier at GBs results in more restriction of carriers. In consequence, n_{off} of 200 nm DNTT film with minimal barrier may be closer to total carrier density (n_{tot}). Therefore, it is more reasonable to use this value as n_{tot} to calculate Debye length. From the estimation with n_{off} of $1 \times 10^{14} \text{ cm}^{-3}$, the Debye length can be determined to be about 200 nm. This value was similar to

some reports in semiconductors with low doping level^{31,34}.

According to your comments, we use the more reasonable assumptions and deductions, and find that the Debye length is smaller than we calculated before. The grain size of 520 nm is larger than $2l_D$, so the barrier of GBs should be discussed in two conditions:

$$E_B = \begin{cases} \frac{q^2 n l^2}{8 \epsilon_s \epsilon_0} & l < 2L_D \\ \frac{q^2 Q_t^2}{8 \epsilon_s \epsilon_0 n} & l > 2L_D \end{cases} \quad (19)$$

When $l < 2l_D$, the $E_B \propto l^2$, as indicated in Eq. (19). As the grain size increases to more than $2l_D$ ($l > 2L_D$), E_B becomes independent of l , corresponding to Eq. (20).

This finding is very important to explain why the proper grain size (corresponding to the proper potential barrier at grain boundary) may cause the largest temperature dependent charge transporting (i.e., the largest sensitivity). The details will be described in the Our reply for Comments 9. Thanks again for your professional and constructive comments and suggestions, which truly help us to understand our results deeply and to find the proper explanation.

Concerning your doubt whether the grain boundary engineering works on the precise tuning of barrier height, there are already some reported works on this topic. (1) John Y. W. Seto proposed a barrier model at grain boundary in 1975³⁵, in which the height of barrier is mainly related to identical grain size l (cm), trap concentration Q_t (cm⁻²) at grain boundary, and doping density N (cm⁻³). (**Fig. R12**). (2) Donghang Yan group investigated surface potential of polycrystalline organic semiconductors by KPFM, and found that GBs barrier heights vary with grain sizes³¹ (**Fig. R13**). However, these studies did not investigate the influence of potential barrier on the temperature dependence of charge transport, which is the key discovery of our work.

Based on these model and method, we achieved the controllable tuning of barrier

height by grain boundary engineering, and investigated the significance effect of potential barrier at GBs on the temperature dependence of charge transport, which is the key discovery in our work, and has not been reported previously.

Fig. R12. The curve of potential barrier at GBs and charge density n , and the maximum of barrier can be obtained when $N =$ trap concentration $Q_t /$ grain size L . (*J. Appl. Phys.* **1975**, 46, 5247-5254.)

Fig. R13. The variation of GBs potentials with grain sizes measured by KPFM. (*Appl.*

Phys. A **2008**, 95, 125-130.)

Our revisions: We modified the estimation of Debye length by recalculated carrier density in the off-state. The corresponding data and discussion were added into main text.

Comments 9: In equations (4)-(5), the author should explain more about the assumptions used here. In a semiconductor textbook, $n(V_g, T) = N_c(T) \exp(-E_B/k_B T)$ is often found for instance a textbook written by Sze, where $N_c(T)$ is temperature variant. Although I am not confident about how this expression is applicable to organic semiconductors, particularly in the thermionic emission model, the authors should take more cares of taking the derivative with T (i.e., dn/dT). It will be appreciated that the authors will communicate with semiconductor physicist to clarify why 26 meV (thermal energy of RT) plays a vital job on maximizing sensitivity.

Our reply: Thanks for your professional comments and kind suggestions, which guides us to think some points deeply and correct some errors. We made a mistake in the analysis of maximum sensitivity in our origin manuscript because of ignoring the dependence of N_c on T . Thanks for pointing out this error!

According to your comments, the knowledge in books and literatures, and the discussions with other experts in semiconductor physics, we found that it is more reasonable to derive the sensitivity (i.e., temperature dependence) by the current density through thermionic emission at GB. The differential sensitivity can be defined as

$$S_D = \frac{1}{J} \frac{dJ}{dT} \quad (21)$$

Inserting Eq (7) $J = A^* T^2 \exp\left(-\frac{E_F - E_V + E_B - qV_B}{k_B T}\right)$, we can find that

$$S_D = \frac{2T + \frac{E_F - E_V + E_B - qV_B}{k_B}}{T^2} \quad (22)$$

Obviously, the sensitivity shows positive correlation with $(E_F - E_V + E_B - qV_B)$, which is consistent with the fitting results (discussed in Our reply for Comment 6). Assuming the consistent dopant density (discussed in Our reply for Comment 8), $E_F - E_V$ makes no difference for devices with the same material. Therefore, the key parameter influencing sensitivity is $E_B - qV_B$.

$E_B - qV_B$ can be defined as the effective height of barrier E_{Be} in bias voltage. As we have discussed, V_B is increased with grain size l , given by Eq. (9). E_B is given by Eq. (19) and (20). Using these Equations together, we obtain

$$E_{Be} = E_B - qV_B = \begin{cases} \frac{q^2 n}{8\epsilon_s \epsilon_0} l^2 - \frac{q\beta V_{DS}}{L} l, & l < 2l_D \\ \frac{q^2 Q_t^2}{8\epsilon_s \epsilon_0 n} - \frac{q\beta V_{DS}}{L} l, & l > 2l_D \end{cases} \quad (23)$$

The Schematic diagram of functional relationship between E_{Be} and l is shown in **Fig. R14**. If $l < 2l_D$, E_{Be} is a quadratic function of l , and decreases firstly and then increases quickly with the increase of l . It should be noted that E_{Be} is negative when l is very small, which means the potential barrier is too small to restrain charge carrier. If $l > 2l_D$, traps have been completely filled, so E_B no longer varies with l . While V_B still increases, E_{Be} thus begins to decrease with l . Therefore, E_{Be} should have a maximum when l is around $2l_D$, which produces highest temperature sensitivity.

Fig. R14. Schematic diagram of the variation of effective height of barrier E_{Be} with grain size l .

In our experiment, l_D is estimated to be 200 nm, and grain size l varies from 200 nm to 520 nm. Sensitivity increases firstly and then decreases. The maximum is obtained when l is 350 nm, which is near to $2l_D$ of about 400 nm. These results are consistent with theory analysis.

Above all, the reason why proper grain size that corresponds to proper potential barrier may yield largest temperature dependence of charge transport (i.e., largest temperature sensitivity) can be understood briefly as follows: According to Eqs. (22), (23) and (24), sensitivity depends on the effective height of barrier E_{Be} , in which the potential barrier E_B and the voltage V_B are competitive and can be tuned by grain size. If the applied voltage is defined value, V_B always increases lineally with grain size. As shown in Figure R14, the E_{Be} - l relationship derived from Eqs. (23) and (24) has two regimes: (1) E_B generally increases as a quadratic function of l when $l < 2l_D$ (Eq. (23)). (2) If $l > 2l_D$, the traps at GBs are completely filled, rendering E_B not to increase with l . However, V_B still enlarges, so E_{Be} begins to decline with l (Eq. (24)). Therefore, maximum of E_{Be} (i.e., sensitivity) should be obtained when l is around $2l_D$, which further causes the strong temperature dependence of charge transport.

Our revisions: We made major revisions to the mechanism analysis and interpretation of modulating temperature dependence. The corresponding data and discussion were added into main text and Supplementary Information.

Below are the experts in semiconductor physics and organic semiconductor who have deeply discussed with us on this work and given us constructive and professional suggestions.

Prof. Zhongming Wei (semiconductor physics)

State Key Laboratory of Superlattices and Microstructures, Institute of Semiconductors, Chinese Academy of Sciences, Beijing 100083, P. R. China.

E-mail: zmwei@semi.ac.cn

Prof. Jun Kang (semiconductor physics)

Beijing Computational Science Research Center, Beijing 100193, P. R. China.

E-mail: jkang@csrc.ac.cn

Dr. Wenchong Wang (semiconductor physics)

Physikalisches Institut, Westfälische Wilhelms-Universität, Wilhelm-Klemm-Strasse 10, 48149 Münster, Germany

E-mail: wangw@uni-muenster.de

Prof. Chengliang Wang (organic semiconductor)

School of Optical and Electronic Information, Huazhong University of Science and Technology, Wuhan, Hubei 430074, P. R. China

E-mail: clwang@hust.edu.cn

Prof. Tao Li (semiconductor physics)

School of Chemistry and Chemical Engineering, Shanghai Jiao Tong University, Shanghai 200240, P. R. China.

E-mail: litaotao1983@sjtu.edu.cn

Prof. Yanyan Fu (semiconductor and sensor)

Shanghai Institute of Microsystem and Information Technology, Chinese Academy of
Sciences, Shanghai 200240, P. R. China.

E-mail: fuyy@mail.sim.ac.cn

Reply for Reviewer #2

Comments 1: The manuscript entitled “Effectively Modulating Thermal Activated Charge Transport in Organic Semiconductors by Precise Potential Barrier Engineering” is focusing on studying the grain size effect on the transfer properties of the DNTT transistors and utilize the grains with different sizes and boundary density to achieve temperature sensor functions. The overall quality of the work is good and it should be published in nature communication. It covers from fundamental science to practical applications. Using the grain size effect to achieve temperature sensing is a very good idea. Here I summarized some of the comments for the authors to further improve the manuscript.

Our reply: We greatly appreciate your high and constructive comments and suggestions, which help us to improve our manuscript, and make some unclear points clear.

Below we would like to address your comments one by one.

Comments 2: Fig. 3(d) is I_{off} , and it maybe confusing if the axis title used I_{DS} .

Our reply: Thanks for your meticulously observation! Under your kind reminder, we double checked the title of all the Figures, and modified the unclear points. According to the comments by Reviewer #1, we modified the data fitting, and Fig. 3d was replaced with Fig. 3e in the revised manuscript.

Our revisions: Some titles in Fig. 3 and 5 were modified. The necessary explains were added.

Comments 3: The extraction of the mobility and carrier concentration in Fig. 3f is a bit confusing to me. Did the author use mobility and conductivity to get the carrier concentration or did they use the carrier concentration and conductivity to get the

mobility? If it is the first case, how did they calculate the mobility? If it is the 2nd case, how did they calculate the carrier concentration?

Our reply: Thank you for pointing out this unclear point! Reviewer #1 also raised the similar question. According to your comments, we gave the calculation method of the saturated region mobility. We evaluated the dependence of mobility on gate voltage, and thus derived the mobility in the off-state, with which the carrier density in the off-state was calculated by drift equation.

Below we would like to explain how to calculate the mobility (μ) and carrier density (n):

We used the field-effect method to calculate the saturated region mobility by

$$I_{ds} = \frac{W}{2L} \mu_{sat} C_i (V_g - V_{th})^2 \quad (1)$$

In order to estimate carrier density in the off-state, the dependence of mobility on gate voltage (**Fig. R1**, shown below) should be considered. DNTT is a thienothiophene small molecule organic semiconductor. These molecules are highly delocalized, closely stacked and a low degree of disorder^{10,11}. The researches by Liu suggest that the $\mu - n$ relation in OFETs is closely related to the degree of delocalization and order of organic semiconductors^{12,13}. **Fig. R1** shows the correlation between the dependency of μ on $V_g - V_{th}$ (i.e., the dependency of μ on n) and the degree of delocalization in organic semiconductors. We also extracted the $\mu - (V_g - V_{th})$ relation from transfer characteristic of the DNTT device as shown in **Fig. R2**, which indicates the weak dependency of μ and V_g . The μ decreases by less than an order of magnitude from on-state ($V_g - V_{th} = 20V$) to off-state ($V_g - V_{th} = 0$). It is similar to TIPS-pentacene with $\Delta D = 0.7$ and $\square = 0.36$ (red line in **Fig. R1 a** and purple triangle in **Fig. R1 b**). We used the μ at $V_g - V_{th} = 0$ (the change of μ is negligible when $V_g < V_{th}$) as the estimated value of μ in the off-state to approximately calculate

charge density n , which would not cause a large deviation in the estimation of n .

Fig. R1. a) The calculated mobility as a function of gate voltage for different ΔD . b) The values of ΔD and ΔE are plotted for different semiconductors. The gray arrow shows the direction of increasing disorder in charge transport. (Chuan Liu et. al *Mater. Horiz.* **2017**, 4, 608)

Fig. R2. The dependence of μ on $(V_g - V_{th})$ of the DNTT OFET.

According to the comments by Reviewer #1, we recalculated carrier density in the off-state by drift equation. The current density is defined as the amount of charge that passes through a unit area per unit time.

$$J = \frac{Q}{St} \quad (2)$$

where Q is quantity of charge, and S is sectional area. Assuming that the length of charge drifting in time t is L , Eq. (2) becomes

$$J = \frac{QL}{SLt} = \frac{QL}{Vt} \quad (3)$$

Inserting $\frac{Q}{V} = qn$ and $\frac{L}{t} = v$ into Eq. (3), it becomes

$$J = qnv \quad (4)$$

Where n is volume density of charge carrier, and v is drift velocity of charge carrier.

Using $v = \mu E$ with Eq. (15), the current density can be given by

$$J = qn\mu E \quad (5)$$

where E is electric field intensity. The carrier density in off-state can be estimated through Eq. (5) with field-effect mobility extract by transfer curve in off-state ($V_g - V_{th} = 0$).

Our revisions: We have recalculated the n_{off} of the DNNT OFETs and update the corresponding data.

Comments 4: The structure of the device used in Fig. 5 should be given. Does it have the same structure as the device used in Fig. 3c? What gate bias is used to measure the δI ?

Our reply: Thank you for pointing out this unclear point! All devices used in this work are organic field effect transistor (OFET), as shown in **Fig. 1b**. In the application for temperature sensing, the gate bias is -20V, the same as used in the discussion of temperature dependence

Our revisions: We have added the explain about the structure and gate bias of the devices in the main text, and added the structure diagram as the insert into Fig. 5a. (Fig. R3)

Fig. R3. Temperature resolution curve of sensor measured at 37-38 °C with a precision of 0.2 °C. The inset shows the structure of the device.

Comments 5: More information about the origin of equation (2) and (3) would be useful.

Our reply: Thanks for your helpful suggestion! According to theoretical model of GBs^{31,35}, if $l < 2l_D$, the crystallite is completely depleted of carriers and the traps are partially filled (i.e., $nl < Q_t$),

$$E_B = q\phi_B = \frac{q^2 l^2 n}{8\epsilon_s \epsilon_0} \quad l < 2l_D \quad (6)$$

If $l > 2l_D$, the crystallite is partly depleted and the traps are fully filled (i.e., $nl > Q_t$), which enable E_B to be independent of l ,

$$E_B = q\phi_B = \frac{q^2 Q_t^2}{8\epsilon_s \epsilon_0 n} \quad l > 2l_D \quad (7)$$

where ϵ_0 and ϵ_s are vacuum permittivity and relative permittivity, respectively.

$l_D = \sqrt{\epsilon_s \epsilon_0 k_B T / q^2 n}$ is the characteristic length of SCR and screen length against the trapped charges at the GBs^{38,39}.

Eqs. (6) and (7) are the Eqs. (2) and (3) in our origin manuscript, which was firstly

given by John Y. W. Seto in 1975³⁵ for describing the height of the barrier at grain boundary (GBs) of polycrystalline silicon. Donghang Yan et al. used these equations to explain the variation of potential at GBs measured by KPFM with grain size in organic semiconductors³¹.

Our revisions: The references and discussions of the origin of equation (2) and (3) were added into the main text.

Reference

1. Ren, X., et al. A Low-Operating-Power and Flexible Active-Matrix Organic-Transistor Temperature-Sensor Array. *Adv. Mater.* **28**, 4832-4838 (2016).
2. C., G. D. Physics: Principles with Applications, 4th ed. Prentice Hall (1995).
3. Hong, S. Y., et al. Stretchable Active Matrix Temperature Sensor Array of Polyaniline Nanofibers for Electronic Skin. *Adv. Mater.* **28**, 930-935 (2016).
4. Kwon, I. W., Son, H. J., Kim, W. Y., Lee, Y. S., Lee, H. C. Thermistor behavior of PEDOT:PSS thin film. *Synth. Met.* **159**, 1174-1177 (2009).
5. Lv, Y., Hu, M., Wu, M., Liu, Z. Preparation of vanadium oxide thin films with high temperature coefficient of resistance by facing targets d.c. reactive sputtering and annealing process. *Surf. Coat. Technol.* **201**, 4969-4972 (2007).
6. Moreno, M., Kosarev, A., Torres, A., Ambrosio, R. Fabrication and performance comparison of planar and sandwich structures of micro-bolometers with Ge thermo-sensing layer. *Thin Solid Films* **515**, 7607-7610 (2007).
7. Bethe, H. A. Theory of the boundary layer of crystal rectifiers. *MIT Radiat Lab. Rep.* 43-12 (1942).
8. Sze, S. M., Ng, K. K. Physics of semiconductor devices. John Wiley & sons (2006).
9. Rhoderick, E. H., Williams, R. Metal-semiconductor contacts, 2nd Ed. Clarendon, Oxford (1988).
10. Illig, S., et al. Reducing dynamic disorder in small-molecule organic semiconductors by suppressing large-amplitude thermal motions. *Nat. Commun.* **7**, 10736 (2016).
11. Peng, B., Huang, S., Zhou, Z., Chan, P. K. L. Solution-Processed Monolayer Organic Crystals for High-Performance Field-Effect Transistors and Ultrasensitive Gas Sensors. *Adv. Funct. Mater.* **27**, 1700999 (2017).
12. Liu, C., et al. A unified understanding of charge transport in organic semiconductors: the importance of attenuated delocalization for the carriers. *Mater. Horiz.* **4**, 608-618 (2017).
13. Yang, T., et al. Understanding, Optimizing, and Utilizing Nonideal Transistors Based on Organic or Organic Hybrid Semiconductors. *Adv. Funct. Mater.* **30**, 1903889 (2019).
14. Bolognesi, A., et al. Effects of Grain Boundaries, Field-Dependent Mobility, and Interface Trap States on the Electrical Characteristics of Pentacene TFT. *IEEE Transactions on Electron Devices* **51**, 1997-2003 (2004).
15. Sirringhaus, H., et al. Two-dimensional charge transport in self-organized, high-mobility conjugated polymers. *Nature* **401**, 685-688 (1999).
16. Li, L., Hu, W., Fuchs, H., Chi, L. Controlling Molecular Packing for Charge Transport in Organic Thin Films. *Adv. Energy Mater.* **1**, 188-193 (2011).
17. Michelotti, F., Bussi, S., Dominici, L., Bertolotti, M., Bao, Z. Space charge effects in polymer-based light-emitting diodes studied by means of a polarization sensitive electroreflectance technique. *J. Appl. Phys.* **91**, 5521-5532 (2002).
18. Hamaguchi, A., et al. Single-Crystal-Like Organic Thin-Film Transistors Fabricated

- from Dinaphtho [2, 3-b: 2', 3'-f] thieno [3, 2-b] thiophene (DNTT) Precursor–Polystyrene Blends. *Adv. Mater.* **27**, 6606-6611 (2015).
19. Mori, H., Takimiya, K. Efficient Photocurrent Generation at Dinaphtho [2, 3-b: 2', 3'-f] thieno [3, 2-b] thiophene/C60 Bilayer Interface. *Appl. phys.express* **4**, 061602 (2011).
 20. Ogawa, Y., Takiguchi, E., Mamada, M., Kumaki, D., Tokito, S., Katagiri, H. Synthesis, crystal structure, and FET characteristics of thieno [2, 3-b] thiophene-based bent-thienoacenes. *Tetrahedron Lett.* **58**, 963-967 (2017).
 21. Zheng, Y. q., Zhang, J. h., Wei, B. Photocurrent enhancement via structural templating of phthalocyanine based planar heterojunction photovoltaics by a thin layer of dinaphthothienothiophene (DNTT) or 3, 4, 9, 10-perylene-tetracarboxylic-dianhydride (PTCDA). *physica status solidi (a)* **212**, 364-368 (2015).
 22. Podzorov, V. Organic single crystals: Addressing the fundamentals of organic electronics. *MRS Bull* **38**, 15-24 (2013).
 23. Podzorov, V., Menard, E., Rogers, J. A., Gershenson, M. Hall effect in the accumulation layers on the surface of organic semiconductors. *Phys. Rev. Lett.* **95**, 226601 (2005).
 24. Uemura, T., et al. Band-like transport in solution-crystallized organic transistors. *Current Applied Physics* **12**, S87-S91 (2012).
 25. Ishii, H., Kobayashi, N., Hirose, K. Strong anisotropy of momentum-relaxation time induced by intermolecular vibrations of single-crystal organic semiconductors. *Phys. Rev. B* **88**, 205208 (2013).
 26. Jacobs, I. E., Moule, A. J. Controlling Molecular Doping in Organic Semiconductors. *Adv. Mater.* **29**, 201703063 (2017).
 27. Yoo, S. J., Kim, J. J. Charge transport in electrically doped amorphous organic semiconductors. *Macromol Rapid Commun.* **36**, 984-1000 (2015).
 28. Chwang, A. B., Frisbie, C. D. Temperature and gate voltage dependent transport across a single organic semiconductor grain boundary. *J. Appl. Phys.* **90**, 1342-1349 (2001).
 29. Kelley, T. W., Frisbie, C. D. Gate voltage dependent resistance of a single organic semiconductor grain boundary. *J. Phys. Chem. B* **105**, 4538-4540 (2001).
 30. Horowitz, G. Tunneling current in polycrystalline organic thin-film transistors. *Adv. Funct. Mater.* **13**, 53-60 (2003).
 31. Huang, H., Wang, H., Zhang, J., Yan, D. Surface potential images of polycrystalline organic semiconductors obtained by Kelvin probe force microscopy. *Appl. Phys. A* **95**, 125-130 (2008).
 32. Horowitz, G., Hajlaoui, M. E., Hajlaoui, R. Temperature and gate voltage dependence of hole mobility in polycrystalline oligothiophene thin film transistors. *J. Appl. Phys.* **87**, 4456-4463 (2000).
 33. Schön, J. H., Kloc, C. Charge transport through a single tetracene grain boundary. *Appl. Phys. Lett.* **78**, 3821-3823 (2001).
 34. Singh, B., Singh, J., Kaur, J., Moudgil, R. K., Tripathi, S. K. Thermally and optical induced effects on the structural and electrical parameters of nc-CdS thin films.

- Journal of Materials Science: Materials in Electronics* **27**, 8701-8709 (2016).
35. Seto, J. Y. W. The electrical properties of polycrystalline silicon films. *J. Appl. Phys.* **46**, 5247-5254 (1975).
 36. Chwang, A. B., Frisbie, C. D. Field effect transport measurements on single grains of sexithiophene: Role of the contacts. *J. Phys. Chem. B* **104**, 12202-12209 (2000).
 37. Weins, M., Gleiter, H., Chalmers, B. Computer Calculations of the Structure and Energy of High-Angle Grain Boundaries. *J. Appl. Phys.* **42**, 2639-2645 (1971).
 38. Riess, I. Conditions for neglecting space charge effects on distributions of point defects and I–V relations. *Solid State Ionics* **69**, 43-52 (1994).
 39. Gil, Y., Umurhan, O., Riess, I. Properties of solid state devices with mobile ionic defects. Part I: The effects of motion, space charge and contact potential in metal| semiconductor| metal devices. *Solid State Ionics* **178**, 1-12 (2007).

Reviewers' Comments:

Reviewer #1:

Remarks to the Author:

I appreciated the authors that they have kindly addressed almost all my comments in the revised manuscript.

However, I do not agree with the publication of this paper. Let me first summarize the authors' achievements:

1. The authors have successfully assessed the anomalous temperature dependence of off-state current of OFETs, and demonstrated reasonably high current sensitivity with respect to temperature.
2. The authors attempted to analyze this anomalous temperature dependence, and concluded that this originates from the grain boundary size relative to the Debye screening length.
3. The authors have successfully demonstrated highly-sensitive temperature sensors.

First, I would say that the authors' finding #1 is novel (to the best of my knowledge, there is no such report in organic electronics community, but I am not 100% sure in a (poly)-silicone electronics).

Second, the authors' conclusion #2 is not fully supported by their experimental results at all, I mean, the argument of Debye length is merely a speculation.

The authors derived the Debye length, that is the most important parameter in this work, with the assumption of carrier concentration of 10^{14} cm⁻³.

This value is really questionable.

The way to derive carrier concentration at off state by the authors is to assume the carrier concentration of on-state as $10^{18}\sim 10^{19}$ cm⁻³, and simply to divide this value by the on-off ratio of 105 to get the carrier concentration of 10^{14} cm⁻³.

I would NEVER accept this simplistic calculation.

In abstract of the revised manuscript, the authors clearly concluded that "this strategy theoretically and experimentally discloses that ... when E_{be} reaches the maximum at the point of grain size near to twice of Debye length...".

However, there is no experimental nor theoretical backup for pinpointing the Debye screening length.

The authors listed some references to justify the value of off-state carrier concentration, but there is no experimental backups either in the listed references.

To the best of my knowledge, approximately 2.5×10^{14} cm⁻³ is the lower limit of carrier concentration that is determined by a trustable transport measurements (See T. Kaji et al, *Advanced Materials*, 21 3689 (2009)).

It should be noted that this value is achieved for a single crystal rubrene with an extremely-high purity.

I do not believe that the carrier concentration of present DNTT system (thermal evaporation on the polymer surface) is better than this value.

In addition, as the authors stated in the revised manuscript, there is a certain amount of trap density-of-states (trap DOS) mainly due to grain boundaries, and the random potentials of CYTOP surface.

Given this trap DOS (in-gap DOS) and the Fermi-Dirac distribution, there should be large amount of carriers thermally-excited in the off-state.

This is why the off-current of the present OFET is relatively high (on the order of subnano A).

Furthermore, the thermionic emission model presented in the manuscript is specialized too much to appeal to a broad audience of *Nature Communications*.

I am afraid to say that I could not catch up the whole twelve equations in the revised manuscript. There must be prerequisite conditions in each equation, and it is impossible for me to judge whether they are all satisfied in OFETs.

Overall, I do not give credit on the argument of Debye length and thermionic emission.

The speculation that appears throughout the revised manuscript worsens the quality of paper.

Last, in terms of the demonstration of temperature sensors (#3), I am afraid to say that this demonstration does not provide a groundbreaking insight on real applications.

The present temperature sensors based on off-state OFETs geometry inherently have extremely high input impedance (>10 MOhm), which is not appreciated in the real analog circuit because the electronic noise level is larger when the resistance is larger.

Overall, I do not agree with the publication of this paper as its present form.

It is necessary to clarify what can be concluded strictly by experimental results and what is the authors' speculation.

Clearly the authors' finding #1 is of interest to a broad audience.

It would be the best to emphasize this aspect.

If I may have a suggestion, it may be convincing if the authors develop the facts as follows;

The high temperature sensitivity demonstrated in off-state OFETs (~ 155) can be far larger than what is expected from a standard thermally-activated carrier transport.

When a typical activation energy of 100 meV for organics is assumed, the expected temperature sensitivity can be around 2 from 298K to 358K.

This comparison will be more interesting and intuitive.

Minor

- In Fig.1c, please display " $V_{ds} = -60$ V" either in Fig.1c or in caption for clarity.

- In Fig.1d, please display " $V_{ds} = -60$ V" and " $V_{gs} = 0$ V?" either in Fig.1d or in caption for clarity.

- In Fig.3a and in the main manuscript, the expression of sensitivity is strange and misleading, it should be $[I(V_g, T) - I(V_g, T_0)] / I(V_g, T_0)$, as I have suggested before. The authors' definition $\Delta I(V_g, \Delta T)$ in the revised manuscript is mathematically strange. Clearly, $I(V_g, T) - I(V_g, T_0)$ and $\Delta I(V_g, \Delta T)$ are not equivalent.

- In Fig.3b, a unit of TCR should be unified, %K⁻¹ is used in the revised manuscript, while %degC⁻¹ is used in Fig.3b.

- In Fig.3f, please define $\Delta\mu_{off}/\mu_{off0}$ and $\Delta n_{off}/n_{off0}$. They should be $[\mu_{off}(T) - \mu_{off}(T=298)] / \mu_{off}(T=298)$, and $[n_{off}(T) - n_{off}(T=298)] / n_{off}(T=298)$. I do not prefer to use the subscription of off0 (off + zero), which is really misleading. Alternatively, the authors should reconsider the plot: not taking the differential, but merely plotting $\mu_{off}(T) / \mu_{off}(T=298)$ and $n_{off}(T) / n_{off}(T=298)$. In this case, the values given at $T=298$ K are unity. This is more intuitive because multiplying $\mu_{off}(T) / \mu_{off}(T=298)$ and $n_{off}(T) / n_{off}(T=298)$ gives the sensitivity value.

- In Fig.5, the resolution is really bad. Please modify this. Also, a unit of T is broken.

- In Fig.5, the authors should double-check the value of sensitivity because the sensitivity was measured to be 155 in Fig.3, but the value in Fig.5 is 4 at 1K difference meaning that 240 at 60 K difference. Please explain this inconsistency.

- In the method section, I would suggest the authors to remove pentacene section to the supplementary information, because all experimental results related to pentacene devices are shown only in SI.

- Regarding Comment 4, I have just suggested to compare the sensitivity with a standard manner ie using TCR. With the current sensitivity of 155, TCR is estimated to be 1.65%K⁻¹. I mean the current sensitivity of 155 and TCR of 1.65%K⁻¹ are equivalent by definition. So I am happy if the author pointed out this (just saying 1.65%K⁻¹ is good enough for a temperature sensor.) I would suggest that the sentences "However, it should be noted... Therefore, the sensitivity..." should be deleted. Also, in SI, the section of "the discussion of TCR" should be deleted because the presented argument is obvious.

This is just to let you know that when we use a temperature sensor in real applications, the output is voltage most likely because these analog signal must be processed with a comparator or amplifier, where the TCR is more useful to design the circuit.

Reviewer #2:

Remarks to the Author:

I think the authors have address most of my concern nicely and I believe the manuscript is ready for publication. Here are my two last minor concerns,

The concept of "averaged TCR" is a bit weird as we all know that TCR value is highly localized temperature dependence and it will induce a lot of error if we use average value. So we usually use TCR value at certain temperatures, such as TCR300K

One last concern is the calculation of the carrier concentration by using the mobility extracted from the off state. Since the off state current of the OFETs is relatively large in the current work, which may lead to an overestimation of n_{off} .

Reviewer #3:

Remarks to the Author:

The authors have addressed my major points in full. The quality of the work has improved and the paper can be published.

Notes: Figure * denotes the Figure in the main text, Figure R* denotes Figure in the reply for reviewer below, and Figure S* denotes Fig. in the Supplementary Information. For clarity and convenience, several Figures appear twice in our reply for different comments

Reply for Reviewer #1

Comments 1: I appreciated the authors that they have kindly addressed almost all my comments in the revised manuscript.

However, I do not agree with the publication of this paper. Let me first summarize the authors' achievements:

1. The authors have successfully assessed the anomalous temperature dependence of off-state current of OFETs, and demonstrated reasonably high current sensitivity with respect to temperature.
2. The authors attempted to analyze this anomalous temperature dependence, and concluded that this originates from the grain boundary size relative to the Debye screening length.
3. The authors have successfully demonstrated highly-sensitive temperature sensors.

Our reply: Thank you very much for reviewing our manuscript very patiently and meticulously. Your constructive comments and kind suggestions in the last and this review process truly guide us to improve the quality (including novelty and significance) of manuscript, correct some errors, and make some unclear points clear. Your summarization of our achievements is very accurate. Thanks again! We have carefully addressed all your comments in this round and made the corresponding revisions.

For clarity and convenience, the key explanations and revisions are listed as follows:

- (1) Some redundant and messy contents in the last reply may make you misunderstand the way we calculated carrier concentration in the off-state. We apologized and clearly clarified it in **Our reply for comment 3**.
- (2) The reason why the carrier concentration in DNTT film is more likely to be lower than organic single crystalline was explained in **Our reply for comment 4**.
- (3) The investigation of anomalous temperature dependence was emphasized, the theoretical model was simplified in the main text, and some parts are moved to the Supplementary Information. (**Our reply for comment 5 and 7**).
- (4) The feasible solution to high impedance of temperature sensors based on off-state OFETs was introduced. (**Our reply for comment 6**).
- (5) The minor errors were corrected. (**Our reply for comment 8**)

Below we would like to address your comments one by one.

Comments 2: First. I would say that the authors' finding #1 is novel (to the best of my knowledge, there is no such report in organic electronics community, but I am not 100% sure in a (poly)-silicone electronics).

Our reply: Thanks a lot for your high comment! We have also tried our best to search for the temperature dependence of output current (including off-state and on-state) of OFETs from Web of Sciences database, and did not find such report in organic electronic community.

We compared the temperature dependence of the DNTT OFETs with the highly sensitive (poly)-silicon devices. According to our investigation, thermistors based on polysilicon normally show $TCR < 1 \%K^{-1}$ at room temperature¹⁻³, and transistors based on polysilicon generally show the relative current change lower than one order of magnitude at the temperature difference of 60 K⁴⁻⁷. Compared to the silicon electronics that we surveyed, our devices show higher temperature dependence.

Our revisions: The find #1 is further highlighted in the revised manuscript according

to this **Comments** and **Comments 7** (providing a comparison with the standard thermally-activated carrier transport). Thanks a lot! One simple sentence is added to address the comparison of this work with (poly)-silicon electronics (Page 12).

Comments 3: Second, the authors' conclusion #2 is not fully supported by their experimental results at all, I mean, the argument of Debye length is merely a speculation.

The authors derived the Debye length, that is the most important parameter in this work, with the assumption of carrier concentration of 10^{14} cm^{-3} .

This value is really questionable.

The way to derive carrier concentration at off state by the authors is to assume the carrier concentration of on-state as $10^{18} \sim 10^{19} \text{ cm}^{-3}$, and simply to divide this value by the on-off ratio of 10^5 to get the carrier concentration of 10^{14} cm^{-3} .

I would NEVER accept this simplistic calculation.

In abstract of the revised manuscript, the authors clearly concluded that “this strategy theoretically and experimentally discloses that ... when E_{Be} reaches the maximum at the point of grain size near to twice of Debye length...”.

However, there is no experimental nor theoretical backup for pinpointing the Debye screening length.

Our reply: We apologize for some redundant and messy contents in the last reply, which may make you misunderstand the way we calculated carrier concentration in the off-state.

In fact, **the carrier concentration in the off-state of 10^{14} cm^{-3} was calculated by drift equation $J = qn_{off}\mu_{off}E$ with our experimental data, and the simple speculation (by the carrier concentration and on/off ratio) just served as an additional evidence to support the calculation with drift equation by experimental data.**

For clarity, we would like to describe the calculation process below:

In the drift equation ($J = qn_{off}\mu_{off}E$), q is elemental charge, $E = V_{ds}/L$ is electric

field intensity, and current density J is obtained from the off-state of transfer curve. Therefore, if μ in the off-state can be gotten by a reasonable way, the n_{off} can be calculated.

DNTT is a thienothiophene small molecule organic semiconductor. These molecules are highly delocalized, closely stacked and a low degree of disorder^{8,9}. The researches by Liu et al. suggest that the organic semiconductors with high degree of delocalization and order show weak dependence of mobility on gate voltage (**Fig. R1**)^{10,11}. The dependence of μ and $(V_g - V_{th})$ in this work was examined. As shown in **Fig. R2** (Supplementary Fig. 5), the result indicates the weak dependency of μ and V_g , i.e., μ decreases by less than an order of magnitude from on-state to off-state. Therefore, n_{off} was calculated by approximating μ at $V_g - V_{th} = 0$ as μ_{off} .

Fig. R1. a) The calculated mobility as a function of gate voltage for different ΔD . b) The values of ΔD and ΔE are plotted for different semiconductors. The gray arrow shows the direction of increasing disorder in charge transport. (Chuan Liu et. al *Mater. Horiz.* **2017**, 4, 608)

Fig. R2. The dependence of μ on $(V_g - V_{th})$ of the DNTT OFET.

We admitted that this extraction of μ_{off} is not absolutely accurate, but it would not yield a large error of the calculation of n_{off} . In fact, there are other methods to characterize the mobility of organic semiconductors such as SCLC and Hall. However, it should be noted that FET characterization is the most reasonable way we can take to estimate μ in the off-state, because (1) SCLC is a common way to test mobility, but it encounters difficulties to measure the mobility along the π -stacking direction of organic field-effect semiconductors. (2) Hall effect measurement is normally suitable for semiconductors with high mobility; otherwise the Hall voltage might be overshadowed by the noise such as thermoelectric voltage and misalignment voltage. Therefore, in organic semiconductors, Hall measurement is always performed on single crystal system with mobility of several to tens of cm^2/Vs .

Our revisions: We modified the contents about the calculation of carrier concentration in the manuscript in order to clearly elucidate the extraction method of carrier concentration in the off-state (Page 14, 15). The detailed information is described in Supplementary section 6 and 7.

Comments 4: The authors listed some references to justify the value of off-state carrier concentration, but there is no experimental backups either in the listed references.

To the best of my knowledge, approximately $2.5 \times 10^{14} \text{ cm}^{-3}$ is the lower limit of carrier concentration that is determined by a trustable transport measurement (See T. Kaji et al, *Advanced Materials*, 21 3689 (2009)).

It should be noted that this value is achieved for a single crystal rubrene with an extremely-high purity.

I do not believe that the carrier concentration of present DNTT system (thermal evaporation on the polymer surface) is better than this value.

In addition, as the authors stated in the revised manuscript, there is a certain amount of trap density-of-states (trap DOS) mainly due to grain boundaries, and the random potentials of CYTOP surface.

Given this trap DOS (in-gap DOS) and the Fermi-Dirac distribution, there should be large amount of carriers thermally-excited in the off-state.

This is why the off-current of the present OFET is relatively high (on the order of subnano A).

Our reply: Thanks for your professional comments! Organic single crystal generally has lower density of impurities and traps than thin film system, so organic single crystal devices should have higher carrier concentration. In fact, only specific impurity may serve as dopant and contribute to formation of free carriers, and most impurities serve as traps or scattering centers in organic semiconductors¹²⁻¹⁶. This point is very different from inorganic semiconductors, which can be vividly demonstrated in **Fig. R3** (from 16th Lecture on Molecular Photoelectric Science by Seth R. Marder). It is still a challenge to justify what kind of impurities could ionize and contribute free carriers in organic semiconductors. Therefore, DNTT film (thermal evaporation on the CYTOP surface) may have more impurities but less carrier concentration than the single crystal rubrene.

Fig. R3. The difference of doping between inorganic and organic materials.

Furthermore, T. Kaji et al¹⁷ also verified the calculation of carrier concentration ($2.5 \times 10^{14} \text{ cm}^{-3}$) and mobility ($0.85 \text{ cm}^2 \cdot \text{V}^{-1} \cdot \text{s}^{-1}$) by the equation $\sigma = nq\mu$ (**Fig. R4**), which method is equivalent with ours. The calculated conductivity ($3.4 \times 10^{-3} \text{ S} \cdot \text{m}^{-1}$) is near to the measured one ($3 \times 10^{-3} \text{ S} \cdot \text{m}^{-1}$) from two-terminal measurements (similar to the off-state measurements for FETs). Ren. X. et al¹⁸ reported pentacene thin film with saturation mobility and conductivity of $1.25 \text{ cm}^2 \cdot \text{V}^{-1} \cdot \text{s}^{-1}$ and $3.503 \times 10^{-4} \text{ S} \cdot \text{m}^{-1}$. With equation ($\sigma = nq\mu$), the carrier concentration can be derived to be about $1.75 \times 10^{13} \text{ cm}^{-3}$, which is also much lower than that of single crystal rubrene. It can be seen that the carrier concentration of organic thin film systems is lower than that of single crystal.

resulting μ and N_A are quite consistent with the conductivity obtained from two-terminal measurements ($3 \times 10^{-3} \text{ S m}^{-1}$) using the equation $\sigma = N_A q\mu$. In addition, the V_{bi} and N_A values

Fig. R4. Screenshot of the reference (*Adv. Mater.* 2009, 21, 3689.) in which T. Kaji et

al. verified the calculation of carrier concentration by the equation $\sigma = nq\mu$.

To the best of our knowledge, there are truly few experimental studies on the calculation of off-state carrier concentration in OFETs. We made a careful calculation with reliable experimental data by drift equation. This calculation is not absolutely accurate, but would not yield a large error. Furthermore, this calculation is suitable to semi-quantitatively or qualitatively interpret the experimental results in this work.

Our revisions: Several sentences were added in the main text (Page 14, 15) in order to (1) clearly demonstrate the way we calculated carrier concentration in the off-state, and (2) describe the fact that the carrier concentration of DNTT film in this work is lower than that of single crystal. More detailed information is described in Supplementary section 7.

Comments 5: Furthermore, the thermionic emission model presented in the manuscript is specialized too much to appeal to a broad audience of Nature Communications.

I am afraid to say that I could not catch up the whole twelve equations in the revised manuscript.

There must be prerequisite conditions in each equation, and it is impossible for me to judge whether they are all satisfied in OFETs.

Overall, I do not give credit on the argument of Debye length and thermionic emission.

The speculation that appears throughout the revised manuscript worsens the quality of paper.

Our reply: Thanks for your constructive comments and suggestions. As you suggested, the anomalous phenomenon of temperature dependence will appeal to a broad audience. And theoretical model of correlation between the temperature dependence and potential barrier may provide reference for devices design, which may be interesting for specific readers. According to your comments and suggestions, we emphasized the investigation of anomalous temperature dependence of OFETs and

simplified the discussion of thermionic emission model in the revised manuscript. The secondary equations were moved to Supplementary Information.

After simplifying the theoretical model, only 6 equations are kept in the main text. We double-checked every equation in the manuscript. Eq. (1) about thermionic emission are widely used in the investigation of Schottky barrier in OFETs¹⁹⁻²¹. Eqs. (2-3), describing potential barrier of grain boundary, were also applied in organic semiconductors²²⁻²⁴, and the prerequisite conditions were assumed in the manuscript. Eqs. (4) and (5) are just a combination of Eqs. (1-3). Eq. (6) is the mathematical transformation of Eq. (1). The rest equations, about TCR and mobility in saturated region, are commonly used in OFETs and sensors. Our applications of the above equations meet the prerequisite conditions of usage.

Our revisions: We emphasized the investigation of anomalous temperature dependence of OFETs and simplified the discussion of thermionic emission model. Some equations were moved to Supplementary Information.

Comments 6: Last, in terms of the demonstration of temperature sensors (#3), I am afraid to say that this demonstration does not provide a groundbreaking insight on real applications.

The present temperature sensors based on off-state OFETs geometry inherently have extremely high input impedance (>10 MOhm), which is not appreciated in the real analog circuit because the electronic noise level is larger when the resistance is larger.

Our reply: Thanks for your professional comment! We agree that the high input impedance of OFET off-state usually leads to large noise. However, the noise originated from the high resistance could be suppressed by circuit design such as integration of low-noise amplifiers²⁵. (**Fig. R5**).

About this issue, we discussed with Prof. Xiaojun Guo (expert in organic semiconductor and circuits) and Dr. Wenchong Wang (expert in semiconductor physics and circuit design). They suggested that such off-state current of OFETs can be read out using commercial charge sensitive pre-amplifier chips with analog-to-digital converter. The chip, owning charge integrator operational amplifier,

can convert charge to voltage output, and has been widely used in active matrix imagers for finger print, X-ray imaging, which have similar level of off current for readout. In addition, the low off-state current is beneficial to reduce static power in practical applications.

The information of two experts

(1) Prof. Xiaojun Guo

School of Electronic, Information and Electrical Engineering, Shanghai Jiaotong University

Email: x.guo@sjtu.edu.cn

(2) Dr. Wenchong Wang

Institute of Physics, Muenster University

wangw@uni-muenster.de

	TBioCAS'11 [30]	TBE'06 [68]	TBioCAS'14 [66]	EMBC'09 [36]	JSSC'14 [38]	JSSC'15 [49]
AE voltage gain	3, 10, 100	100	10	11	11, 51, 101	140, 700, 1200
Supply voltage	1.8V	5V	3V	3.3V	1.8V	1.8V
Input referred noise (per channel)	1.2 μ Vrms (0.5-100Hz)	7.49 μ Vrms (1-1kHz)	0.56 μ Vrms (0.5-100Hz)	2.4 μ Vrms (0.5-100Hz)	1.75 μ Vrms (0.5-100Hz)	0.65 μ Vrms (0.5-100Hz)
Electrode offset tolerance	Rail-to-rail	\pm 250mV	--	Rail-to-rail	\pm 250mV	\pm 350mV
DC-coupling feature	AC-coupling	AC-coupling	AC-coupling	AC-coupling	AC-coupling	"Functionally" DC-coupling
Input impedance	100M Ω @50Hz	1T Ω @DC	100M@50Hz	--	400M Ω @50Hz	100M Ω @50Hz
CMRR @50Hz	82dB (with CMFB)	78dB	64dB	90dB	84dB (with CMFF)	102dB (with CMFF)
AFE power consumption (per channel)	20 μ W (AE only)	7.5mW (AE only)	360 μ W (AE only)	600 μ W incl. ADC	82 μ W incl. ADC	105 μ W incl. ADC
ADC	no	no	no	16 bits	12 bits	12 bits
Number of wires	4	2	4	--	6	5

Fig. R5. Screenshot of the reference (Active Electrodes for Wearable EEG Acquisition: Review and Electronics Design Methodology. *IEEE T. Bio-Med. Eng.* 2017, 10, 187–198) in which authors reviewed the design of electrodes with built-in readout circuitry for wearable devices with high input impedance.

Our revisions: Several sentences are added to address the problem of high input impedance of OFET-based sensors and the feasible solutions (Page 18 in main text).

Comments 7: Overall, I do not agree with the publication of this paper as its present form. It is necessary to clarify what can be concluded strictly by experimental results and what is the authors' speculation.

Clearly the authors' finding #1 is of interest to a broad audience. It would be the best

to emphasize this aspect.

If I may have a suggestion, it may be convincing if the authors develop the facts as follows;

The high temperature sensitivity demonstrated in off-state OFETs (~155) can be far larger than what is expected from a standard thermally-activated carrier transport. When a typical activation energy of 100 meV for organics is assumed, the expected temperature sensitivity can be around 2 from 298K to 358K.

This comparison will be more interesting and intuitive.

Our reply: Thank you very much for the valuable and professional suggestions! In addition, we apologized again for the misleading information about the calculation of carrier concentration, which has been clarified in **Our reply for comment 3**

We clarified the following points are concluded or calculated strictly by our experimental data. For example, 1) the variation tendency of sensitivity with grain sizes and corresponding potential barrier; 2) the calculation of carrier concentration and Debye length; 3) the analyzation of the effective height of the potential barrier and its impact with sensitivity; 4) the demonstration of the temperature sensors. In addition, some approximation and speculations were also made, such as 1) the negligence of tunneling current; 2) simplification of the voltage drop across boundaries; 3) calculation of mobility in the off-state.

Furthermore, in the revised manuscript we emphasized the high temperature dependence of our devices by comparison with the standard thermally-activated carrier transport according to your constructive and kind suggestions.

Our revisions: According to your constructive and kind suggestions, we made some revisions as follows: (1) modified the main text in order to emphasize the high temperature dependence of our devices, and the theoretical discussion part was simplified in the main text. (2) The sentences your suggested were added in the abstract (Page 2) and main text (Page 11) “Significantly, through this strategy a traditional thermo-stable organic semiconductor (dinaphtho[2,3-b:2',3'-f]thieno[3,2-b]thiophene, DNTT) achieves extremely high

thermo-sensitivity (relative current change) of 155, which is far larger than what is expected from a standard thermally-activated carrier transport (Assuming the typical activation energy of 100 -200 meV for organics^{8,18}, the expected temperature sensitivity is only around 1-3 from 298K to 358K).” (3) some approximation and speculations are briefly described in the main text and Supplementary Information.

Comments 8: Minor

- (1) In Fig.1c, please display “ $V_{ds} = -60 \text{ V}$ ” either in Fig.1c or in caption for clarity. In Fig.1d, please display “ $V_{ds} = -60 \text{ V}$ ” and “ $V_{gs} = 0 \text{ V}$?” either in Fig.1d or in caption for clarity.
- (2) In Fig.3a and in the main manuscript, the expression of sensitivity is strange and misleading, it should be $[I(V_g, T) - I(V_g, T_0)] / I(V_g, T_0)$, as I have suggested before. The authors’ definition $\Delta I(V_g, \Delta T)$ in the revised manuscript is mathematically strange. Clearly, $I(V_g, T) - I(V_g, T_0)$ and $\Delta I(V_g, \Delta T)$ are not equivalent.
- (3) In Fig.3b, a unit of TCR should be unified, $\%K^{-1}$ is used in the revised manuscript, while $\%degC^{-1}$ is used in Fig.3b.
- (4) In Fig.3f, please define $\Delta\mu_{off}/\mu_{off 0}$ and $\Delta n_{off}/n_{off 0}$. They should be $[\mu_{off}(T) - \mu_{off}(T=298)] / \mu_{off}(T=298)$, and $[n_{off}(T) - n_{off}(T=298)] / n_{off}(T=298)$. I do not prefer to use the subscription of off0 (off + zero), which is really misleading. Alternatively, the authors should re-consider the plot: not taking the differential, but merely plotting $\mu_{off}(T) / \mu_{off}(T=298)$ and $n_{off}(T) / n_{off}(T)$. In this case, the values given at T=298 K are unity. This is more intuitive because multiplying $\mu_{off}(T) / \mu_{off}(T=298)$ and $n_{off}(T) / n_{off}(T)$ gives the sensitivity value.
- (5) In Fig.5, the resolution is really bad. Please modify this. Also, a unit of T is broken.
- (6) In Fig.5, the authors should double-check the value of sensitivity because the sensitivity was measured to be 155 in Fig.3, but the value in Fig.5 is 4 at 1K difference meaning that 240 at 60 K difference. Please explain this inconsistency.
- (7) In the method section, I would suggest the authors to remove pentacene section to the supplementary information, because all experimental results related to pentacene

devices are shown only in SI.

(8) Regarding Comment 4, I have just suggested to compare the sensitivity with a standard manner ie using TCR. With the current sensitivity of 155, TCR is estimated to be $1.65\%K^{-1}$. I mean the current sensitivity of 155 and TCR of $1.65\%K^{-1}$ are equivalent by definition. So I am happy if the author pointed out this (just saying $1.65\%K^{-1}$ is good enough for a temperature sensor.) I would suggest that the sentences “However, it should be noted... Therefore, the sensitivity...” should be deleted. Also, in SI, the section of “the discussion of TCR” should be deleted because the presented argument is obvious. This is just to let you know that when we use a temperature sensor in real applications, the output is voltage most likely because these analog signals must be processed with a comparator or amplifier, where the TCR is more useful to design the circuit.

Our reply: Thank you very much for your meticulous observation and kind suggestions, according to which we have made the corresponding revisions.

Our revisions: For clarity, we added number (1-8) to the above comments and made the corresponding revisions as follows:

- (1) we displayed “ $V_{ds} = -60\text{ V}$ ” in the caption of Fig. 1c, and “ $V_{ds} = -60\text{ V}$ ” and “ $V_g = 0\text{ V}$ ” in the caption of Fig. 1d.
- (2) We modified sensitivity as $[I(V_g, T) - I(V_g, T_0)]/ I(V_g, T_0)$, and displayed the corresponding value of V_g and T in the caption.
- (3) The unites of TCR were unified as $\%K^{-1}$.
- (4) The titles of vertical axis in Fig. 3f were changed to $[\mu_{off}(T) - \mu_{off}(T=298)]/ \mu_{off}(T=298)$, and $[n_{off}(T) - n_{off}(T=298)]/ n_{off}(T=298)$.
- (5) We replaced Fig. 5 with a high-resolution photograph.
- (6) In Fig. 5a, I/I_0 is 3.8 while $\Delta I/I_0$ is 2.8 at 1K difference meaning that 168 at 60 K difference. It is a little larger than 155, which is within the measurement error.
- (7) The method section about pentacene was moved into supplementary information.
- (8) We deleted the needless discussion about TCR both in main text and SI.

Reply for Reviewer #2

Comments 1: I think the authors have address most of my concern nicely and I believe the manuscript is ready for publication. Here are my two last minor concerns,

Our reply: Thank you very much for your high comments and support!

Comment 2: The concept of “averaged TCR” is a bit weird as we all know that TCR value is highly localized temperature dependence and it will induce a lot of error if we use average value. So we usually use TCR value at certain temperatures, such as TCR300K

Our reply: Thanks for your high and professional comments! According to your comment, we deleted the definition of average value of TCR in Supplementary Information.

Our revisions: The define of average value of TCR was deleted.

Comments 3: One last concern is the calculation of the carrier concentration by using the mobility extracted from the off state. Since the off-state current of the OFETs is relatively large in the current work, which may lead to an overestimation of n_{off} .

Our reply: Thanks for your professional comments! The off-state current of our OFETs is indeed relatively large, so we checked the gate current. However, no discernible leakage of current was found (as shown in **Fig. R5**). Therefore, the off-state current factually reflects the free carrier concentration of the DNTT film.

Fig. R5. The characteristic curves of the DNTT OFET with grain size of 200 nm.

Reply for Reviewer #3

Comments 1: The authors have addressed my major points in full. The quality of the work has improved and the paper can be published.

Our reply: Thank you very much for your high comments and support! Your comments and suggestions help us improve the quality of the manuscript.

Reference

1. Paivi Heimala, P. K., Jaakko Aarnio, and Arja Heinamaki. Thermally Tunable Integrated Optical Ring Resonator with Poly-Si Thermistor. *J. Lightwave Technol.* **14**, 2260 (1996).
2. Poyai, J. S. E. R. C. H. A. Characteristics of silicon thin film thermistors. *Proceedings of ECTI-CON*, 2008.
3. Zhizhen Wu, C. L., Jed Hartings, Sthitodhi Ghosh, Raj Narayan and Chong Ahn. Polysilicon-based flexible temperature sensor for brain monitoring with high spatial resolution. *J. Micromech Microeng* **27**, 025001 (2017).
4. Hashim, Y., Sidek, O. Effect of temperature on the characteristics of silicon nanowire transistor. *J. Nanosci. Nanotechnol.* **12**, 7849-7852 (2012).
5. Hashim, Y., Sidek, O. Simulation study of temperature sensitivity of silicon nanowire transistors with different types of orientations. *IEEJ Transactions on Electrical and Electronic Engineering* **7**, 458-460 (2012).
6. Lei, W.-D., Bai, Y., Tian, H., Zhao, F. A Single Device Temperature Sensor Based on

- Amorphous Silicon Thin Film Transistor. *IEEE Sensors Journal* **19**, 10236-10242 (2019).
7. Shimanovich, K., Mutsafi, Z., Roizin, Y., Rosenwaks, Y. CMOS compatible SOI nanowire FET with charged dielectric for temperature sensing applications. *J. Phys. D Appl. Phys.* **53**, 065101 (2020).
 8. Ren, X., et al. A Low-Operating-Power and Flexible Active-Matrix Organic-Transistor Temperature-Sensor Array. *Adv. Mater.* **28**, 4832-4838 (2016).
 9. C., G. D. Physics: Principles with Applications, 4th ed. Prentice Hall (1995).
 10. Hong, S. Y., et al. Stretchable Active Matrix Temperature Sensor Array of Polyaniline Nanofibers for Electronic Skin. *Adv. Mater.* **28**, 930-935 (2016).
 11. Kwon, I. W., Son, H. J., Kim, W. Y., Lee, Y. S., Lee, H. C. Thermistor behavior of PEDOT: PSS thin film. *Synth. Met.* **159**, 1174-1177 (2009).
 12. Curtin, I. J., Blaylock, D. W., Holmes, R. J. Role of impurities in determining the exciton diffusion length in organic semiconductors. *Appl. Phys. Lett.* **108**, 163301 (2016).
 13. Jurchescu, O. D., Baas, J., Palstra, T. T. M. Effect of impurities on the mobility of single crystal pentacene. *Appl. Phys. Lett.* **84**, 3061-3063 (2004).
 14. Li, Y., et al. In situ purification to eliminate the influence of impurities in solution-processed organic crystals for transistor arrays. *J. Mater. Chem. C* **1**, 1352-1358 (2013).
 15. Street, R. A., Chabinyo, M. L., Endicott, F. Chemical impurity effects on transport in polymer transistors. *Phys. Rev. B* **76**, 045208 (2007).
 16. Urien, M., et al. Field-effect transistors based on poly(3-hexylthiophene): Effect of impurities. *Org. Electron.* **8**, 727-734 (2007).
 17. Kajji, T., Takenobu, T., Morpurgo, A. F., Iwasa, Y. Organic Single-Crystal Schottky Gate Transistors. *Adv. Mater.* **21**, 3689-3693 (2009).
 18. Ren, X., Chan, P. K. L., Lu, J., Huang, B., Leung, D. C. W. High Dynamic Range Organic Temperature Sensor. *Adv. Mater.* **25**, 1291-1295 (2013).
 19. Horowitz, G., Hajlaoui, M. E., Hajlaoui, R. Temperature and gate voltage dependence of hole mobility in polycrystalline oligothiophene thin film transistors. *J. Appl. Phys.* **87**, 4456-4463 (2000).
 20. Ruden, P. P., Smith, D. L. Theory of spin injection into conjugated organic semiconductors. *J. Appl. Phys.* **95**, 4898-4904 (2004).
 21. Yunus, M., Ruden, P. P., Smith, D. L. Ambipolar electrical spin injection and spin transport in organic semiconductors. *J. Appl. Phys.* **103**, 103714 (2008).
 22. Huang, H., Wang, H., Zhang, J., Yan, D. Surface potential images of polycrystalline organic semiconductors obtained by Kelvin probe force microscopy. *Appl. Phys. A* **95**, 125-130 (2008).
 23. Scheller, L. P., Nickel, N. H. Charge transport in polycrystalline silicon thin-films on glass substrates. *J. Appl. Phys.* **112**, 013713 (2012).
 24. Verlaak, S., Arkhipov, V., Heremans, P. Modeling of transport in polycrystalline organic semiconductor films. *Appl. Phys. Lett.* **82**, 745-747 (2003).
 25. Xu, J., Mitra, S., Van Hoof, C., Yazicioglu, R. F., Makinwa, K. A. A. Active Electrodes for Wearable EEG Acquisition: Review and Electronics Design Methodology. *IEEE*

Rev. Biomed. Eng. **10**, 187-198 (2017).

Reviewers' Comments:

Reviewer #1:

Remarks to the Author:

I appreciate again the authors that they have carefully considered my comments in the revised manuscript.

I am now happy with most of their reply.

However, I am still largely negative to this paper, particularly about the argument of carrier concentration, which is a critical factor to justify the theoretical model proposed by the authors.

The drift equation was used to derive the carrier concentration.

The authors evaluated the mobility at the off-state from the saturation regime of FET.

Clearly, the drift equation holds as long as the electric field is uniform across a conductive media, whereas in the saturation regime in FET the electric field is not uniform at all.

It is obvious that the current increases linearly with E in the drift equation, but the current saturates with E in the saturation regime of FET.

These two equation should not be used at the same time.

This internal inconsistency worsens the validity of their carrier concentration.

In the ref. 18 (in the reply letter), the value of carrier concentration was not specifically evaluated by Ren and coworkers. So I do not believe that this reference supports 10^{14} cm^{-3} carrier concentration.

I strongly suggest to tone down the argument of the Debye screening length throughout the manuscript.

I do not accept the exaggerated sentences, for example in the Abstract, "This strategy discloses that"

At the same time, I feel that the paper may lose its impact and significance if the argument of the theoretical model is eliminated from the manuscript.

Overall, I am still very negative to this paper.

Reviewer #2:

Remarks to the Author:

I believe the authors have addressed all the concerns and comments raised by all the reviewers.

This manuscript is ready for publication in Nature Communications.

Reply for Reviewer #1

Comments 1: I appreciate again the authors that they have carefully considered my comments in the revised manuscript.

I am now happy with most of their reply.

However, I am still largely negative to this paper, particularly about the argument of carrier concentration, which is a critical factor to justify the theoretical model proposed by the authors.

The drift equation was used to derive the carrier concentration.

The authors evaluated the mobility at the off-state from the saturation regime of FET.

Clearly, the drift equation holds as long as the electric field is uniform across a conductive media, whereas in the saturation regime in FET the electric field is not uniform at all.

It is obvious that the current increases linearly with E in the drift equation, but the current saturates with E in the saturation regime of FET.

These two equations should not be used at the same time.

This internal inconsistency worsens the validity of their carrier concentration.

Our reply: Thank you for reviewing our manuscript for three times although the comments in this round are critical and negative. We highly appreciated your comments during the past review rounds, which truly helps us deeply understand our work and constructively improve the manuscript.

Please further spare some time to read our explanation for the internal consistence between drift equation and saturation equation in FET as well as the validity of carrier concentration calculation. Thanks for your patience and support!

As stated in these comments, in our work the drift equation ($J=qn\mu E$, Eq. 1) was used to derive the carrier concentration, and the mobility in the off-state was

evaluated from the saturation regime of FET with the standard equation ($I_D = \frac{W}{2L} C_{ox} \mu (V_G - V_T)^2$, Eq. 2). In fact, these two equations have internal consistence because of the following points:

(1) **Classical literatures¹⁻⁴ and textbooks⁵⁻⁷ demonstrate that the equation in the saturation regime of FET (Eq. 2) is derived from the drift equation (Eq. 1), and Eq. 1 is still valid when electric field is non-uniform.** In fact, the drift equation is a classical equation to describe the motion of carriers under electric field, and has a wide range of applications, even for some devices with complex potential distribution. Most of equations, that describe the motion of carriers driven by electric field, are derived from drift equation. **Figure R1** shows the screenshot of textbook-level reviews of FET (*Chem. Mater.* 2004, 16, 4436-4451, times cited: 1091. *Chem. Rev.* 2007, 107, 4, 1296–1323, times cited: 1651), which demonstrates the following points: I-V characteristics in linear and saturation regime of FET are derived from the drift equation, and uniform electric field is not the necessary for drift equation, so the drift equation is still valid in the saturation regime of FET although the electrical field is not uniform.

Figure R1. I-V characteristics in linear and saturation regime of FET are derived from drift equation. (screenshot of the textbook-level literatures). Note: Eq. (3) in left panel and Eq. 2.3 in right panel are the different forms of drift equation, and can be transformed to the standard form.

(2) **Drift equation is still valid when the current saturates with electric field (E) in the saturation regime of FET.** In fact, in the saturation regime of FET, the electric field at the drain extends the depletion region and narrows the effective conductive channel (“pinch-off” effect), which causes the carrier concentration decreasing from the source to the drain electrode (**Figure R2a**)¹. In another word, **in the saturation regime, the electric field increases, but the average carrier concentration induced by gate voltage in the channel decreases, which leads to the current saturation effect (Figure R2b).** Therefore, the saturation effect does not invalidate the drift equation. Otherwise, the Eq. 2 could not be derived from Eq. 1. Many classical books⁵⁻⁷ and literatures¹⁻⁴ can support this point.

Figure R2. a) Schematic structure of FETs in saturation regime (screenshot of the reference, *Chem. Rev.* 2007, 107, 4, 1296–1323.). b) Illustration of the validity of the drift equation in saturation regime.

Above all, it can be found that **the Eq. 1 is the basis of Eq. 2 and the saturation effect totally accords with the drift equation**, so there is no internal inconsistency between these two equations at the basic physical level, which may guarantee the validity of the calculation of carrier concentration.

In addition, we want to address another point as follows: since discovery, organic electronics have undergone significant progresses especially at the materials and manufacture, but the device physics and models are still immature mainly because organic systems are very complex and such investigation is highly challenging. This work studied the thermal-activated charge transporting of organic semiconductors. We found anomalous and strong temperature-dependent charge transporting in FET and established a model to interpret it. The novelty and significance of this work have also been recognized by you and another two reviewers. Based on the above “Our reply for Comments 1” and the description in the manuscript, we believe that our model can explain the experimental results. Moreover, our experimental observation and theoretical model may provide meaningful information for the field of organic electronics. We sincerely hope you can support us. Your support will be another motivation for us to continue working on organic electronics especially at the aspect of device physics. Thank you very much for your understanding and supporting!

Our revisions: The brief description about the internal consistence between drift equation and saturation equation is added into the main text (Page 14) to demonstrate the validity of carrier concentration calculation.

Comments 2: In the ref. 18 (in the reply letter), the value of carrier concentration was not specifically evaluated by Ren and coworkers. So I do not believe that this reference supports 10^{14} cm^{-3} carrier concentration.

I strongly suggest to tone down the argument of the Debye screening length throughout the manuscript.

I do not accept the exaggerated sentences, for example in the Abstract, “This strategy discloses that”

At the same time, I feel that the paper may lose its impact and significance if the argument of the theoretical model is eliminated from the manuscript.

Overall, I am still very negative to this paper.

Our reply: It is often reported⁸⁻¹³ that the organic semiconductors with μ of around 1 show conductivity σ of 10^{-4} to $10^{-6} \text{ S cm}^{-1}$, so the carrier concentration n can be simply calculated to be 10^{14} - 10^{12} cm^{-3} by $\sigma = nq\mu$. Though Ren and coworkers⁸ do not evaluate specifically, n can also be simply derived to be in the above range. The literature you listed also used this equation and reported the value in this range¹³. Our calculation is well in the reported reasonable range.

In this work, we reported the anomalous and strong temperature dependence of OFETs, and further proposed a theoretical model to analyze the experimental results. According to “**Our reply for Comments 1**”, **the calculation of carrier concentration with drift equation is valid, so theoretical model for the experimental results should be valid** as well.

On the other hand, the anomalous temperature dependence is one key significance and novelty of this work, as you commented in the last review round. Therefore,

according to your suggestions in the last review round, we have already toned down the theoretical model part. However, if we eliminate the theoretical part, it would be difficult for the potential readers to understand this work. In fact, we cherished and respected all your comments and suggestions during the past three review rounds, but could not accept your suggestions here. Sorry for this point! Thanks for your understanding and supporting!

Reply for Reviewer #2

Comments 1: I believe the authors have addressed all the concerns and comments raised by all the reviewers. This manuscript is ready for publication in Nature Communications.

Our reply: Thank you very much for your high comments and support! Your comments and suggestions help us improve the quality of the manuscript.

Reference

1. Zaumseil, J., Sirringhaus, H. Electron and Ambipolar Transport in Organic Field-Effect Transistors. *Chem. Rev.* **107**, 1296–1323 (2007).
2. Newman, C. R., Frisbie, C. D., da Silva Filho, D. A., Brédas, J. L., Ewbank, P. C., Mann, K. R. Introduction to Organic Thin Film Transistors and Design of n-Channel Organic Semiconductors. *Chem. Mater.* **16**, 4436-4451 (2004).
3. C. T. Sah, Characteristics of the metal-Oxide-semiconductor transistors. *IEEE Transactions on Electron Devices.* **11**, 324-345 (1964).
4. Ihantola, H. K. J., Moll, J. L. Design theory of a surface field-effect transistor. *Solid State Electron.* **7**, 423-430 (1964).
5. Sze, S. M., Ng, K. K. *Physics of semiconductor devices*. John wiley & sons, 2006.
6. Neamen, D. A., Ed. *Semiconductor Physics and Devices: Basic Principles*. 3rd ed.; Irwin: Chicago, 2002.
7. Sze, S. M. *Semiconductor Devices Physics and Technology*. John Wiley and Sons, Inc.: New York, 2002.
8. Ren, X., Chan, P. K. L., Lu, J., Huang, B., Leung, D. C. W. High Dynamic Range Organic Temperature Sensor. *Adv. Mater.* **25**, 1291-1295 (2013).
9. Nollau, A., Pfeiffer, M., Fritz, T., Leo, K. Controlled n-type doping of a molecular organic

- semiconductor: Naphthalenetetracarboxylic dianhydride (NTCDA) doped with bis(ethylenedithio)-tetrathiafulvalene (BEDT-TTF). *Journal of Applied Physics* **87**, 4340-4343 (2000).
10. Chiang, C. K., et al. Electrical Conductivity in Doped Polyacetylene. *Physical Review Letters* **39**, 1098-1101 (1977).
 11. Kleemann, H., Schuenemann, C., Zakhidov, A. A., Riede, M., Lüsse, B., Leo, K. Structural phase transition in pentacene caused by molecular doping and its effect on charge carrier mobility. *Organic Electronics* **13**, 58-65 (2012).
 12. Walzer, K., Maennig, B., Pfeiffer, M., Leo, K. Highly Efficient Organic Devices Based on Electrically Doped Transport Layers. *Chemical Reviews* **107**, 1233-1271 (2007).
 13. Kaji, T., Takenobu, T., Morpurgo, A. F., Iwasa, Y. Organic Single-Crystal Schottky Gate Transistors. *Adv. Mater.* **21**, 3689-3693 (2009).

Reviewers' Comments:

Reviewer #1:

Remarks to the Author:

I am grateful that the authors responded to my comments.

Although I am still skeptical of estimation of carrier number, I agree with the acceptance of this paper.

However, I feel that the argument about the Debye length should be left as an open question.

I meant to say that the authors did assume the drift equation to a simple Ohms law; the current linearly responds to an applied voltage.

But no evidence of the linear response of IV characteristics has been provided.

It would be great if the authors provide I_{ds} vs V_{ds} characteristics of off-state in FETs, and summarize all the parameters to estimate carrier number (I_{ds} , V_{ds} , L , W , d ; thickness, and mobility) in supplementary information.

Now, the authors used the mobility at $V_g=0$, for which a certain number of carrier is accumulated at the channel because this condition does not satisfy the flat-band condition.

It means that the mobility used here is apparently the upper limit.

I would strongly suggest the authors to discuss the error bar in carrier density, by considering the lower limit of mobility.

In addition, a simultaneous use of drift equation and FET's saturation equation may lead another issue; the former describes a purely bulk conduction, and the later does an interfacial, 2D conduction.

The author should leave this issue as an open question, and clearly mention that the present method to estimate the Debye length is not established.

Reply for Reviewer #1

Comments 1: I am grateful that the authors responded to my comments.

Although I am still skeptical of estimation of carrier number, I agree with the acceptance of this paper.

However, I feel that the argument about the Debye length should be left as an open question.

Our reply: Thank you very much for your understanding, support and constructive and kind suggestions! Indeed, the precise estimation of the Debye length in organic semiconductors is very difficult due to the uncertain concentration of dopants. We admitted that our method may be not absolutely accurate although we have tried our best to give a reasonable method to calculate Debye length. According to your comments and suggestions, we clearly pointed out the difficulty of the Debye length calculation in the manuscript and left it as an open question. Thanks again!

Our revisions: The corresponding discussion was added into the main text (Page 16).

Comments 2: I meant to say that the authors did assume the drift equation to a simple Ohm's law; the current linearly responds to an applied voltage.

But no evidence of the linear response of IV characteristics has been provided.

It would be great if the authors provide I_{ds} vs V_{ds} characteristics of off-state in FETs, and summarize all the parameters to estimate carrier number (I_{ds} , V_{ds} , L , W , d ; thickness, and mobility) in supplementary information.

Our reply: Thanks for your professional suggestions! According to them, we checked the I-V characteristic of the device by two-terminal measurement. As shown in **Figure R1**, the device exhibits linear I-V characteristic. Furthermore, we summarized all the parameters to estimate carrier concentration in the **Table R1**. Furthermore, we counted the parameters of 8 OFET devices in different batches, I_{ds} and mobility in the off-state are in the range of $1.36 \times 10^{-8} - 4.98 \times 10^{-9}$ A and $0.35 - 0.13 \text{ cm}^2 \cdot \text{V}^{-1} \cdot \text{s}^{-1}$,

respectively. Therefore, the carrier concentration can be calculated to be $9.8 \times 10^{13} - 1.45 \times 10^{14} \text{ cm}^{-3}$ with these parameters.

Figure R1. I-V characteristic of the device by two-terminal measurement.

Table R1. Summary of the parameters to estimate carrier concentration.

V_{ds} (V)	I_{ds} (A)	L (um)	W (um)	d (nm)	μ_{off} ($\text{cm}^2 \cdot \text{V}^{-1} \cdot \text{s}^{-1}$)
-60	$1.36 \times 10^{-8} - 4.98 \times 10^{-9}$	50	1000	20	0.35 – 0.13

Our revisions: We added **Figure R1** (I-V characteristics at $V_g = 0 \text{ V}$) and **Table R1** and some discussion into Supplementary Information (Supplementary Figure S5 and Table S2).

Comments 3: Now, the authors used the mobility at $V_g=0$, for which a certain number of carrier is accumulated at the channel because this condition does not satisfy the flat-band condition.

It means that the mobility used here is apparently the upper limit.

I would strongly suggest the authors to discuss the error bar in carrier density, by considering the lower limit of mobility.

Our reply: Thanks for your valuable and constructive suggestion. We admitted that the approximation of the mobility in the off-state is not absolute strictness, but it is enough to qualitatively explain the anomalous temperature dependence in this work.

Furthermore, it is noted that the estimation of the lower mobility limit is quite difficult or impossible to some extent. To our knowledge, there are no reports on the off-state mobility in organic semiconductors.

On the other hand, our OFETs show turn-on voltage higher than 0 V (more negative voltage), which means that the flat band condition is not satisfied at $V_g = 0$ V. The channel has few carrier accumulation before the flat band condition is satisfied, so the mobility at $V_g = 0$ V should be very close to that in the off-state.

According to your suggestions, we tried another way to give the error bar of the carrier concentration. We calculated the current density and mobility of the 8 devices in different batches in the off-state. And the error bar of carrier concentration can be calculated to be $9.8 \times 10^{13} - 1.45 \times 10^{14} \text{ cm}^{-3}$ with these parameters (**Our reply for comment 2**).

Our revision: We added the discussion of error bar into the main text and Supplementary Information by considering the devices in different batches (Page 15 in main text and Page 13 in Supplementary Information).

Comments 4: In addition, a simultaneous use of drift equation and FET's saturation equation may lead another issue; the former describes a purely bulk conduction, and the later does an interfacial, 2D conduction.

The author should leave this issue as an open question, and clearly mention that the present method to estimate the Debye length is not established.

Our reply: Thanks for your kind and constructive comments and suggestions. As your comments, the saturation equation of FET normally describes 2D conduction in the on-state, which makes it difficult to accurately calculate the mobility in the off-state. Consequently, we tried reasonable approximations as much as possible, i.e., approximating mobility at $V_g = 0$ V to that in off-state. We acknowledge it is not absolutely accurate, but it is in the reasonable range and is enough to qualitatively explain the anomalous temperature dependence in this work (discussed in **Our reply**

for comment 3). As you suggested, it would be more precise to leave this issue as an open question, and clearly mention that the present method to estimate the Debye length is not established.

Our revisions: we have modified the statements about estimation of Debye length (mainly in the combination of saturation equation in FET and drift equation), and left it as an open question. Thanks again!